# Zeolite-promoted platinum catalyst for efficient reduction of nitrogen oxides with hydrogen

Shaohua Xie [1,2,6], Liping Liu [3,6], Yuejin Li[4], Kailong Ye[1,2], Daekun Kim[2], Xing Zhang[2], Hongliang Xin [3] ✉, Lu Ma[5], Steven N. Ehrlich[5] & Fudong Liu [1,2] ✉

Internal combustion engine fueled by carbon-free hydrogen ($H_2$-ICE) offers a promising alternative for sustainable transportation. Herein, we report a facile and universal strategy through the physical mixing of Pt catalyst with zeolites to significantly improve the catalytic performance in the selective catalytic reduction of nitrogen oxides ($NO_x$) with $H_2$ ($H_2$-SCR), a process aiming at $NO_x$ removal from $H_2$-ICE. Via the physical mixing of $Pt/TiO_2$ with Y zeolite ($Pt/TiO_2$ + Y), a remarkable enhancement of $NO_x$ reduction activity and $N_2$ selectivity was simultaneously achieved. The incorporation of Y zeolite effectively captured the in-situ generated water, fostering a water-rich environment surrounding the Pt active sites. This environment weakened the NO adsorption while concurrently promoting the $H_2$ activation, leading to the strikingly elevated $H_2$-SCR activity and $N_2$ selectivity on $Pt/TiO_2$ + Y catalyst. This study provides a unique, easy and sustainable physical mixing approach to achieve proficient heterogeneous catalysis for environmental applications.

The transportation sector has a considerable impact on global climate change[1], being responsible for nearly 24% of the world's $CO_2$ emissions stemming from fossil fuel combustion[2]. Consequently, it is imperative to prioritize substantial $CO_2$ reduction within this sector. While electric powertrains powered by renewable energy hold promise, their environmental cost and limited energy capacity for heavy-duty vehicles pose significant challenges for widespread application[3]. There is another viable avenue lies in the adoption of internal combustion engines (ICE) operating on carbon-free hydrogen ($H_2$), which presents a promising alternative for sustainable transportation[3,4]. During the $H_2$ combustion process, nitrogen oxides ($NO_x$) are the primary environmental pollutants[5,6]. Selective catalytic reduction (SCR) of $NO_x$ is one of the most efficient and widely used technologies for $NO_x$ abatement in excess oxygen[6,7]. For $H_2$-ICE applications, $H_2$ extracted from the fuel tank can serve directly as a reducing agent for the SCR of $NO_x$

($H_2$-SCR)[8]. This approach may offer significant economic and environmental benefits. However, to make this technique viable, the key issue to be solved is the development of robust $H_2$-SCR catalyst systems, which can demonstrate excellent low-temperature $NO_x$ reduction activity and $N_2$ selectivity simultaneously.

Supported platinum (Pt) and palladium (Pd) catalysts have been extensively investigated for $H_2$-SCR reaction[6,9,10]. Notably, Pt catalysts have shown great promise with their superior low-temperature (<150 °C) activity comparing to Pd catalysts[11,12], although there is urgent need for significant improvement in $N_2$ selectivity[13]. $H_2$ activation was considered as one of the most critical factors on Pt catalysts that could profoundly influence the $H_2$-SCR performance[14]. Improving $H_2$ activation and sustaining abundant *H species on Pt catalysts could positively promote the NO dissociation[15–17], which has been reported as the rate-determining step for the $H_2$-SCR reaction[18,19]. Additionally, this

[1]Department of Chemical and Environmental Engineering, Bourns College of Engineering, Center for Environmental Research and Technology (CE-CERT), Materials Science and Engineering (MSE) Program, University of California, Riverside, CA, USA. [2]Department of Civil, Environmental, and Construction Engineering, Catalysis Cluster for Renewable Energy and Chemical Transformations (REACT), NanoScience Technology Center (NSTC), University of Central Florida, Orlando, FL, USA. [3]Department of Chemical Engineering, Virginia Polytechnic Institute and State University, Blacksburg, VA, USA. [4]BASF Environmental Catalyst and Metal Solutions, Iselin, NJ, USA. [5]National Synchrotron Light Source II (NSLS-II), Brookhaven National Laboratory, Upton, New York, NY, USA. [6]These authors contributed equally: Shaohua Xie, Liping Liu. ✉e-mail: hxin@vt.edu; fudong.liu@ucr.edu; lfd1982@gmail.com

enhancement could also facilitate the formation of $NH_x$ species, which, in some cases, have been found beneficial for the H$_2$-SCR reaction[14,20–23]. Currently, substantial efforts have been dedicated towards increasing the presence of metallic Pt species[24,25], as it plays a crucial role in H$_2$ activation. Studies have reported that specific additives could substantially enhance the $NO_x$ reduction activity by reducing the Pt valence. For instance, the addition of Mo and Na to Pt/SiO$_2$[26] and the introduction of Ti species into Pt/MCM-41[13] resulted in the lowered Pt valence state, leading to widened temperature window for $NO_x$ conversion. Additionally, the acidity or basicity of supports also strongly influenced the dispersion and chemical state of Pt[27,28], with acidic supports being beneficial for the formation of metallic Pt species therefore promoting the H$_2$-SCR performance[27,29]. Such strategies involving the chemical modification of Pt catalysts to form more metallic Pt species were mainly intent to enhance the H$_2$ activation. However, it was observed that the presence of metallic Pt species usually favored the NO adsorption over H$_2$ adsorption, inevitably resulting in a reduced *H coverage during H$_2$-SCR reaction[30]. Moreover, these chemical modification strategies were found to be effective only for specific Pt catalyst systems, and in most cases the enhancement was only restricted to $NO_x$ reduction activity but not to N$_2$ selectivity. Therefore, there is urgent need to design a simple, effective and universal strategy to boost the H$_2$ activation while reducing the NO adsorption on Pt active sites, thus improving the low-temperature activity and N$_2$ selectivity in the H$_2$-SCR reaction on Pt-based catalysts accordingly.

Different from the sophisticated chemical modification strategies as previously reported, in this work, we successfully developed a simple, sustainable physical mixing strategy of oxide-supported Pt catalysts (e.g., Pt/TiO$_2$, Pt/Al$_2$O$_3$, or Pt/SiO$_2$) with various zeolites (e.g., H-Y, H-ZSM-5, H-chabazite (CHA), H-ferrierite (FER), or H-Beta) to significantly promote the H$_2$-SCR reaction. Using this facile approach that is easy to scale up in industry, a universal increase in both the H$_2$-SCR activity and N$_2$ selectivity was achieved. Focusing on a typical physically mixed catalyst system involving the extensively studied Pt/TiO$_2$[31,32] and commercial H-Y zeolite (i.e., Pt/TiO$_2$ + Y), in-depth mechanistic studies were performed through the combined experimental and theoretical approaches. It was clearly revealed that the introduction of Y zeolite facilitated the formation of water-enriched micro-environment on Pt/TiO$_2$, which played a crucial role in

mitigating the over-strong adsorption of NO while promoting the H$_2$ activation on Pt sites. As a result, the disassociation of NO, a crucial step in the H$_2$-SCR reaction, was substantially promoted, leading to the drastic enhancement in the catalytic performance.

## Results

### Physical mixing of Pt catalysts and zeolites to promote the H$_2$-SCR reaction

The Pt/TiO$_2$ catalyst was prepared via a conventional incipient wetness impregnation (IWI) method using colloidal Pt precursor and a commercial TiO$_2$ support. In the H$_2$-SCR reaction under typical given condition, the Pt/TiO$_2$ catalyst showed $NO_x$ conversion above 11% (Fig. 1a) and N$_2$ selectivity above 17% (Fig. 1b) below 250 °C. When physically mixing the Pt/TiO$_2$ catalyst with an inactive commercial H-Y zeolite (SiO$_2$/Al$_2$O$_3$ molar ratio = 30) (Fig. 1a, b), within the investigated temperature range, the Pt/TiO$_2$ + Y catalyst system showed substantially improved catalytic performance, with $NO_x$ conversion above 59% and N$_2$ selectivity above 58% below 250 °C. In addition, this Pt/TiO$_2$ + Y catalyst showed much higher reaction rates and N$_2$ selectivity at 100 and 200 °C compared to most reported Pt and Pd catalysts (Supplementary Table 1). Such a broad operation temperature window (100–250 °C) and excellent catalytic performance from Pt/TiO$_2$ + Y system are highly desirable for the practical H$_2$-SCR application[33]. In the presence of both H$_2$ and O$_2$, NO can either be reduced by H$_2$ to form N$_2$/N$_2$O or be oxidized by O$_2$ to form NO$_2$ (Supplementary Fig. 1). Therefore, it is reasonable that the $NO_x$ conversion and N$_2$ selectivity could hardly achieve 100% under the high space velocity H$_2$-SCR testing conditions with H$_2$O and CO$_2$ (500 ppm NO, 1% H$_2$, 10% O$_2$, 5% CO$_2$, and 5% H$_2$O; WHSV = 461,540 mL·g$_{Pt/TiO2}^{-1}$·h$^{-1}$). Comparing to Pt/TiO$_2$, the Pt/TiO$_2$ + Y system consistently showed higher selectivity towards NO reduction and lower selectivity towards NO oxidation during the H$_2$-SCR reaction (Fig. 1c), particularly at high temperatures. The results clearly demonstrated that the presence of Y significantly promoted the NO reduction by H$_2$ on Pt/TiO$_2$ + Y system.

To verify if there was synergy effect and how it worked between Pt/TiO$_2$ and Y components, we investigated the different physical mixing methods (Supplementary Fig. 2a) and see how the H$_2$-SCR performance was impacted. It was demonstrated that, in clear contrast to the similar catalytic performance (i.e., low $NO_x$ conversion and low N$_2$ selectivity) obtained on Pt/TiO$_2$ + Y-front and Pt/TiO$_2$ + Y-rear

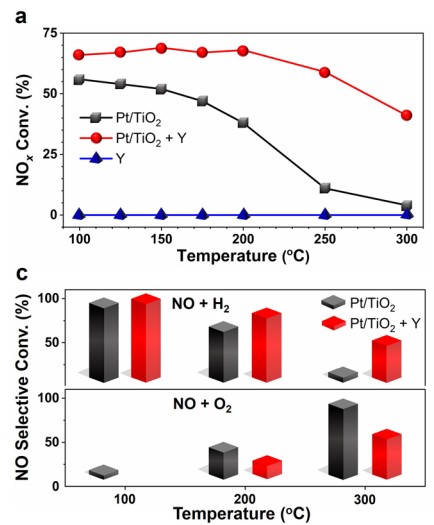
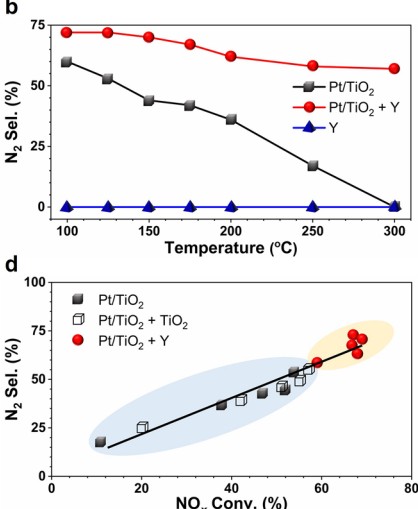

**Fig. 1 | Effect of physical mixing Pt/TiO$_2$ with Y zeolite on the H$_2$-SCR performance. a** $NO_x$ conversion, and (**b**) N$_2$ selectivity in H$_2$-SCR reaction; **c** NO selective conversion (i.e., NO reacting with H$_2$ or O$_2$) in H$_2$-SCR reaction over Pt/TiO$_2$ and Pt/TiO$_2$ + Y catalysts (see Methods section for detailed calculation); **d** Correlation between N$_2$ selectivity and $NO_x$ conversion in H$_2$-SCR reaction over Pt/TiO$_2$,

Pt/TiO$_2$ + TiO$_2$, and Pt/TiO$_2$ + Y catalysts. Reaction conditions: 26 mg of Pt/TiO$_2$ catalyst, or a physical mixture containing 26 mg of Pt/TiO$_2$ and 26 mg of Y or TiO$_2$; steady-state testing; 500 ppm NO, 1% H$_2$, 10% O$_2$, 5% CO$_2$, and 5% H$_2$O; weight hourly space velocity (WHSV) = 461,540 mL·g$_{Pt/TiO2}^{-1}$·h$^{-1}$.

systems, much more excellent $H_2$-SCR performance was achieved on the $Pt/TiO_2 + Y$ system, where $Pt/TiO_2$ and Y powders were thoroughly physically mixed with appropriate contact (Supplementary Fig. 2b, c). To achieve even closer contact between $Pt/TiO_2$ and Y, we further physically mixed the $Pt/TiO_2$ and Y powders in the presence of water, referred as $(Pt/TiO_2 + Y)\_H_2O$, and loaded Pt onto a pre-prepared 50% $TiO_2/Y$ support (denoted as $Pt/TiO_2/Y$). It was observed that the $(Pt/TiO_2 + Y)\_H_2O$ catalyst showed slightly higher activity (Supplementary Fig. 3), and the $Pt/TiO_2/Y$ catalyst exhibited lower activity compared to the $Pt/TiO_2 + Y$ catalyst. However, both catalysts demonstrated lower $N_2$ selectivity than $Pt/TiO_2 + Y$ catalyst. These results evidently suggest the critical role of establishing an appropriate contact between $Pt/TiO_2$ and Y zeolite in enhancing the overall $H_2$-SCR performance. As shown in Supplementary Fig. 4, the optimal content of Y in the $Pt/TiO_2 + Y$ mixture system was determined as 50 wt%, and this formulation was simply denoted as $Pt/TiO_2 + Y$ thereafter. To better understand this system, we also physically mixed the $Pt/TiO_2$ catalyst with additional $TiO_2$, and the $Pt/Y$ catalyst (prepared by IWI method) with additional $TiO_2$ or Y, and used them as reference catalysts. As presented in Supplementary Fig. 5, the physical mixing of $Pt/TiO_2$ and $TiO_2$ showed no obvious impact on the $NO_x$ conversion and $N_2$ selectivity. However, the physical mixing of $Pt/Y$ with $TiO_2$ or Y resulted in considerable enhancement of the $H_2$-SCR performance. It was worth noting that the $Pt/TiO_2 + Y$ formulation outperformed all other catalysts in terms of $H_2$-SCR activity and showed reasonable $N_2$ selectivity. To gain a deeper insight into the Y promotion effect, the relationship between $N_2$ selectivity and $NO_x$ conversion in the $H_2$-SCR reaction on selected catalysts was established, as depicted in Fig. 1d. Interestingly, the $N_2$ selectivity versus $NO_x$ conversion on all catalysts adhered to the same linear relationship, suggesting that the addition of Y or $TiO_2$ did not alter the overall $H_2$-SCR reaction mechanism on $Pt/TiO_2$ catalyst (yet the Y addition might have changed the $N_2$ formation pathway leading to lower $N_2O$ production, which can be verified by the subsequent experimental results and theoretical calculations).

In addition to Y, the use of other types of zeolites for physical mixing with $Pt/TiO_2$ has also been explored in the $H_2$-SCR reaction (Supplementary Fig. 6). Remarkably, the incorporation of different zeolites such as ZSM-5, CHA, FER, and Beta also yielded substantial benefit, significantly enhancing the $H_2$-SCR performance. Considering both the $H_2$-SCR activity and $N_2$ selectivity in the investigated temperature range, it is evident that Y stands out as the optimal zeolite for promoting the $Pt/TiO_2$ catalyst. To simulate the status of catalysts for $H_2$-ICE exhaust purification after prolonged operation, hydrothermal aging on $Pt/TiO_2$ and $Pt/TiO_2 + Y$ catalysts was conducted at 650 °C for 50 h under 10% $H_2O$ and 10% $O_2$. As shown in Supplementary Fig. 7, not only before but also after the hydrothermal aging, the inclusion of Y in $Pt/TiO_2 + Y$ system consistently exhibited remarkable enhancement on the $H_2$-SCR performance, with notably higher $NO_x$ conversion and $N_2$ selectivity achieved than those by the zeolite-free $Pt/TiO_2$ catalyst. To further verify the universality of this physical mixing strategy, the $H_2$-SCR testing on the hydrothermally aged $Pt/Al_2O_3$ and $Pt/SiO_2$ catalysts with and without Y addition was also performed, and the results are shown in Supplementary Fig. 8. Evidently, the aged $Pt/Al_2O_3 + Y$ and $Pt/SiO_2 + Y$ systems demonstrated significantly enhanced activity and $N_2$ selectivity across the entire spectrum of reaction temperatures when contrasted with their Y-absent counterparts. It is clear that physically mixing the conventional Pt/oxide catalysts with zeolites represents a simple yet universally effective strategy for boosting the $H_2$-SCR performance, particularly tailored for the efficient $NO_x$ removal from vehicle exhaust at low temperatures.

## Structural characterization of $Pt/TiO_2$ before and after physical mixing with Y zeolite

It might be expected that the physical mixing with Y could have modified the physicochemical properties of $Pt/TiO_2$ leading to the distinguishable catalytic performance. We excluded this hypothesis by systematically characterizing the $Pt/TiO_2$ and $Pt/TiO_2 + Y$ catalysts using multiple techniques. X-ray diffraction (XRD) (Supplementary Fig. 9) and $N_2$ adsorption-desorption experiments (Supplementary Fig. 10 and Supplementary Table 2) revealed that the physical mixing showed negligible impact on the crystal structure and textual properties including surface area and porosity of both $Pt/TiO_2$ and Y. It was observed that the $Pt/TiO_2 + Y$ system exhibited a similar $TiO_2$ grain size (20.6 nm) to that of $Pt/TiO_2$ (20.0 nm), and its surface area (349 $m^2/g$) and total pore volume (0.318 $cm^3/g$) were approximately the mathematical average of the values for $Pt/TiO_2$ (81 $m^2/g$, 0.178 $cm^3/g$) and Y (709 $m^2/g$, 0.513 $cm^3/g$), respectively. Additionally, the $Pt/TiO_2 + Y$ system demonstrated structural stability, with no apparent changes in crystal structure or textural properties after reaction at 300 °C under testing conditions with $H_2O$. In addition to the presence of micropores with the average diameter of 0.6 nm, Y zeolite also displayed significant mesopore defects with the average diameter of 3.8 nm that were probably formed during the dealumination process for Y zeolite production. These defects could potentially offer a substantial number of special Brønsted acidic sites (i.e., hydroxyls associated to extra-framework Al enriched on the inner pore surface), which might play a crucial role in facilitating the adsorption of $H_2O$ molecules to occupy the mesopore structures[34]. The change of the $H_2O$ adsorption behavior induced by Y zeolite might have altered the $H_2$-SCR reaction pathway on $Pt/TiO_2$, which will be thoroughly discussed in later sections.

As expected, the Pt particles within both $Pt/TiO_2$ and $Pt/TiO_2 + Y$ catalysts showed very similar average sizes (6.0 nm vs. 6.2 nm), Pt dispersions (8.9% vs. 8.5%), CO adsorption features on Pt particle, and Pt-Pt coordination numbers (11.4 vs. 10.7), as evidenced by the characterization results of transmission electron microscopy (TEM), CO pulse titration, in situ diffuse reflectance infrared Fourier transform spectroscopy (DRIFTS) of CO adsorption, and X-ray absorption spectroscopy (XAS) (Fig. 2a, b, Supplementary Figs. 11, 12 and 13, Supplementary Table 3). Furthermore, the linear combination fitting results of X-ray absorption near-edge structure (XANES) for Pt $L_3$-edge demonstrated that the averaged oxidation states of Pt were 0.21 and 0.48 in $Pt/TiO_2$ before and after the Y addition. These values closely resembled the metallic Pt, a finding further supported by the X-ray photoelectron spectroscopy (XPS) analysis of Pt $4d$ (Supplementary Fig. 14, Supplementary Table 4). These results clearly demonstrated that the physical mixing with Y zeolite did not change the structure of $Pt/TiO_2$, and this conclusion was further supported by the observation of almost identical $H_2$ temperature-programed reduction ($H_2$-TPR) profiles on $Pt/TiO_2$ and $Pt/TiO_2 + Y$ (Supplementary Fig. 15). Additionally, the energy dispersive spectroscopy (EDS) mapping results of $Pt/TiO_2 + Y$ revealed that the $Pt/TiO_2$ components were surrounded by Y zeolite particles, without obvious direct interaction between Pt species and Y, before and after $H_2$-SCR reaction (Fig. 2c, Supplementary Fig. 16). Therefore, different from the chemical modifications as reported previously[24–29], the substantial enhancement in $H_2$-SCR performance on $Pt/TiO_2$ by physically mixing with Y was unequivocally attributable to the factors other than the active site modification.

## Understanding on the Y promotion effect in $Pt/TiO_2 + Y$ system

To determine the role of each reactant and the promotion effect of Y, the reaction orders of NO, $H_2$ and $O_2$ were measured for the $H_2$-SCR reaction. It was found that the reaction orders of NO on both $Pt/TiO_2$ and $Pt/TiO_2 + Y$ were dependent on the NO partial pressure (Supplementary Fig. 17a). On $Pt/TiO_2$, at NO partial pressure below 25.3 Pa, the NO reaction order was determined as 0.95, while this value decreased to 0.63 at NO partial pressure above 25.3 Pa. Meanwhile, the reaction orders of $H_2$ and $O_2$ on $Pt/TiO_2$ were determined as 0.48 and −0.13 (Supplementary Fig. 17b, c), respectively. After physical mixing with Y, there was no evident change in the $O_2$ reaction order on $Pt/TiO_2 + Y$ (only from −0.13 to −0.08). However, a notable increase in the NO

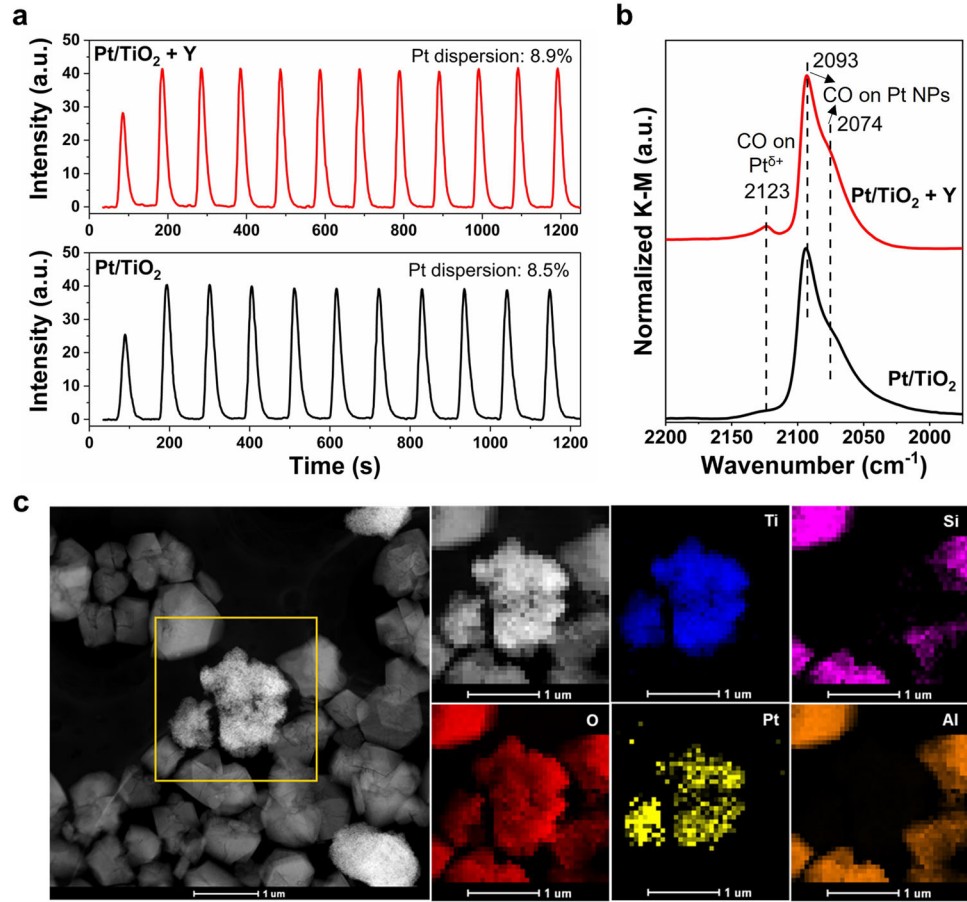

**Fig. 2 | Structural characterization of Pt/TiO₂ + Y. a** CO pulse titration results (with the Pt metal dispersion data inserted); **b** in situ DRIFTS of CO adsorption at 25 °C on Pt/TiO₂ and Pt/TiO₂ + Y catalysts; **c** EDS mapping images for Pt/TiO₂ + Y system.

reaction order (from 0.95 to 1.09 at lower NO partial pressure, and from 0.63 to 0.80 at higher NO partial pressure) and an obvious decrease in the H₂ reaction order (from 0.48 to 0.32) were observed on Pt/TiO₂ + Y. These results suggest that the introduction of Y probably decreased the NO adsorption, and at the same time promoted the H₂ activation because of the potential increase in H₂ coverage on catalyst surface. The enhanced H₂ activation was further supported by the experimental findings presented in Supplementary Figs. 17d and 18. The Pt/TiO₂ + Y system demonstrated superior H₂ oxidation activity compared to the Pt/TiO₂ reference, with the H₂ oxidation activity promoted further as the Y content in the Pt/TiO₂ + Y system increased (Supplementary Fig. 18). This additional increase in H₂ activation could improve the low-temperature activity and reduce the high-temperature activity, as observed on Pt/TiO₂ + Y-67% compared to that on Pt/TiO₂ + Y-50% (Supplementary Fig. 4)[14,25]. Under our testing conditions (500 ppm NO, 1% H₂, and 10% O₂), the H₂-SCR reaction rates can be expressed as: $r_{(Pt/TiO2)} = k_1 \cdot [NO]^{0.63} \cdot [H_2]^{0.48} \cdot [O_2]^{-0.13}$ and $r_{(Pt/TiO2 +Y)} = k_2 \cdot [NO]^{0.80} \cdot [H_2]^{0.32} \cdot [O_2]^{-0.08}$, where $k_1$ and $k_2$ are constants. Notably, the NO and H₂ reaction orders on both Pt/TiO₂ (0.63 and 0.48, respectively) and Pt/TiO₂ + Y (0.80 and 0.32, respectively) are lower than 1. This suggests that the H₂-SCR reaction on both catalysts involved adsorbed NO and dissociated H* species, following the Langmuir-Hinshelwood (L-H) mechanism. Without changing the L-H mechanism, the enhanced H₂ activation could contribute to the improved H₂-SCR activity of Pt/TiO₂ + Y.

The effect of the possibly present NO₂ or NH₃ in the reaction atmosphere on H₂-SCR activity was also studied. In separate NO oxidation testing, the Pt/TiO₂ + Y system displayed noticeably lower NO oxidation activity comparing to Pt/TiO₂ (Supplementary Fig. 19a), and

the presence of NO₂ in the H₂-SCR reaction atmosphere drastically decreased the low-temperature NO$_x$ conversion on both Pt/TiO₂ and Pt/TiO₂ + Y (Supplementary Fig. 19b). Therefore, the presence of any NO₂, generated during H₂-SCR, was not responsible for the enhanced H₂-SCR activity on Pt/TiO₂ + Y. In addition, the potential promotional effect of NH₃ (possibly formed in situ through the reduction of NO$_x$ by H₂) on H₂-SCR activity was ruled out on both Pt/TiO₂ and Pt/TiO₂ + Y. This was evident from the decrease in the H₂-SCR activity observed upon the introduction of NH₃ into the reaction stream at different temperatures (Supplementary Fig. 20).

To understand the impact of Y addition on NO adsorption behavior, the in situ DRIFTS of NO desorption at different temperatures and NO-temperature programmed desorption (NO-TPD) were conducted on Pt/TiO₂ and Pt/TiO₂ + Y catalysts. As shown in Fig. 3a, the NO adsorption on Pt/TiO₂ at 100 °C showed three distinctive bands, corresponding to bridging nitrates (1618 cm⁻¹), bidentate nitrates (1586 cm⁻¹), and monodentate nitrates (1521 cm⁻¹)[35,36]. In clear contrast, the NO adsorption on Pt/TiO₂ + Y exhibited a significantly lower intensity, with the disappearance of monodentate nitrates (Fig. 3b). As the temperature elevated, the nitrate species on both Pt/TiO₂ and Pt/TiO₂ + Y catalysts decreased in intensity, following the sequence of monodentate nitrates > bidentate nitrates > bridging nitrates (Fig. 3a and b). An initial upswing in the bridging nitrates on Pt/TiO₂ was noted, attributed to the intrinsic transformation within the different types of nitrate species[37]. To assess the NO adsorption affinity, the normalized intensities of bidentate nitrates were presented at different temperatures (Fig. 3c). A much more rapid nitrate desorption from Pt/TiO₂ + Y was observed comparing to that from Pt/TiO₂. Such results unequivocally demonstrated the substantial inhibitory effect of Y zeolite on

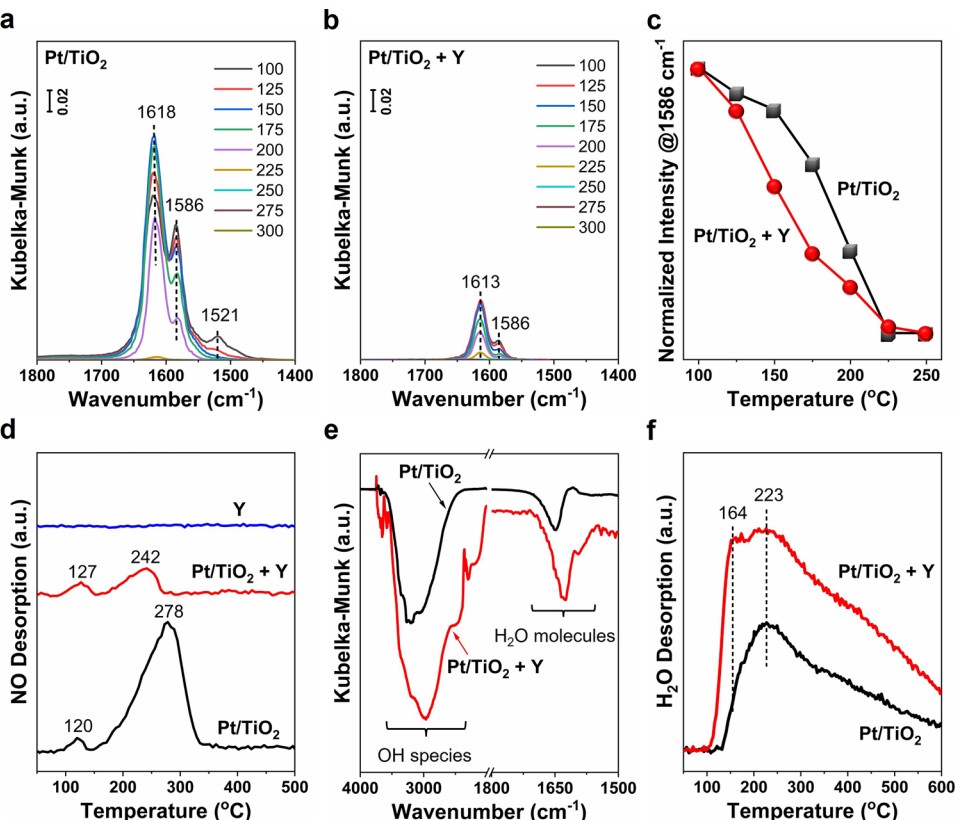

**Fig. 3 | Effects of Y addition on NO and $H_2O$ adsorption properties.** In situ DRIFTS of NO desorption on (**a**) $Pt/TiO_2$ and (**b**) $Pt/TiO_2$ + Y catalysts, and (**c**) normalized peak intensity (at 1586 $cm^{-1}$) for NO adsorption on $Pt/TiO_2$ and $Pt/TiO_2$ + Y catalysts at different temperatures; (**d**) NO-TPD profiles, (**e**) in situ DRIFTS of $H_2O$ adsorption at 120 °C, and (**f**) $H_2O$-TPD profiles on $Pt/TiO_2$ and $Pt/TiO_2$ + Y catalysts.

the NO adsorption onto $Pt/TiO_2$, concurrently fostering the desorption of NO from $Pt/TiO_2$ + Y. These findings were further supported by the NO-TPD results (Fig. 3d), revealing that the $Pt/TiO_2$ + Y system indeed exhibited notably reduced NO desorption intensity and lowered desorption temperature (242 °C) comparing to $Pt/TiO_2$ (278 °C).

Considering that $H_2O$ is the primary product in the $H_2$-SCR reaction, the impact of Y addition on $H_2O$ adsorption property was also investigated. The in situ DRIFTS of $H_2O$ adsorption on both $Pt/TiO_2$ and $Pt/TiO_2$ + Y at 120 °C clearly showed distinct peaks at *ca.* 1630 $cm^{-1}$, indicative of adsorbed $H_2O$ molecules[38]. Additionally, broad peaks at *ca.* 3200 $cm^{-1}$ were observed, corresponding to the hydroxyl species derived from adsorbed $H_2O$ with bending feature[38–40]. Comparing to the case on $Pt/TiO_2$, $H_2O$ adsorption on $Pt/TiO_2$ + Y displayed more prominent peaks (Fig. 3e), suggesting the enhanced $H_2O$ adsorption due to the presence of Y. This enhancement was also confirmed by the $H_2O$-TPD results (Fig. 3f), where more pronounced $H_2O$ desorption peaks were observed on $Pt/TiO_2$ + Y. Other than the $H_2O$ desorption peak observed at 223 °C on both catalysts, an additional desorption peak at 164 °C was detected only on $Pt/TiO_2$ + Y. This low-temperature peak could be attributed to the physically adsorbed $H_2O$ on the Y zeolite.

To reveal the effect of $H_2O$ adsorption on the $H_2$-SCR performance, the catalysts were either pre-dehydrated or pre-adsorbed with $H_2O$ prior to the $H_2$-SCR testing. Under the testing condition without $H_2O$, $Pt/TiO_2$ + Y always outperformed $Pt/TiO_2$ in terms of $NO_x$ conversion and $N_2$ selectivity (Fig. 4a, Supplementary Fig. 21a). Comparing to the situation with pre-dehydration, interestingly, the pre-adsorption of $H_2O$ on both catalysts improved their $H_2$-SCR performance. This improvement was particularly significant regarding the $NO_x$ conversion on $Pt/TiO_2$ catalyst, which exhibited relatively weaker $H_2O$ adsorption capacity as confirmed earlier. The gas phase $H_2O$ formation

was monitored during the $H_2$-SCR reaction (Supplementary Fig. 21b). As expected, much faster increase in $H_2O$ concentration was observed over both catalysts subjected to the pre-adsorption of $H_2O$ comparing to those subjected to the pre-dehydration. It was noticeable that, as shown in Fig. 4b, a discernible correlation emerged between the elevation in $NO_x$ conversion and the concurrent rise in gas phase $H_2O$ concentration over $Pt/TiO_2$ catalyst. However, over $Pt/TiO_2$ + Y system, the rise in gas phase $H_2O$ concentration exhibited a delay compared to the progression of $NO_x$ conversion, suggesting the capture of in situ formed $H_2O$ due to the presence of Y. To further confirm the promotion effect of in situ generated $H_2O$ and to verify the effect of NO adsorption on the $H_2$-SCR activity, transient $H_2$-SCR testing was conducted at 100 °C (Fig. 4c). Using the $NO_x$ concentrations when switching from Ar flow to $H_2$-SCR flow as baselines, significant decrease in $NO_x$ concentrations was observed when switching from $H_2 + O_2$ flow to $H_2$-SCR flow, while obvious increase in $NO_x$ concentrations was observed when switching from $NO + O_2$ flow to $H_2$-SCR flow, on both catalysts. Clearly, initiating a pre-flow of $H_2 + O_2$ yielded benefit on improving the $H_2$-SCR activity, while pre-flowing $NO + O_2$ inhibited the $H_2$-SCR reaction to a certain extent. Such inhibition caused by the $NO + O_2$ flow could be due to the extensive coverage of Pt sites by NO, impeding the activation of $H_2$[30]. At 100 °C, the complete oxidation of $H_2$ to $H_2O$ could be achieved on both catalysts (Supplementary Fig. 18). Consequently, the benefit of pre-flowing $H_2 + O_2$ should be originated from the adsorption of in situ formed $H_2O$. The presence of $H_2O$ could strongly inhibit the NO adsorption, as confirmed by the in situ DRIFTS (Fig. 4d) and NO-TPD (Fig. 4e) analyses conducted on $Pt/TiO_2$ catalyst. The physical mixing of $Pt/TiO_2$ with Y could further promote the adsorption of in situ generated $H_2O$ (Fig. 3e, f), creating a $H_2O$-rich environment around the Pt sites and facilitating the formation of a $H_2O$-covered Pt surface. This surface could reduce the NO coverage on

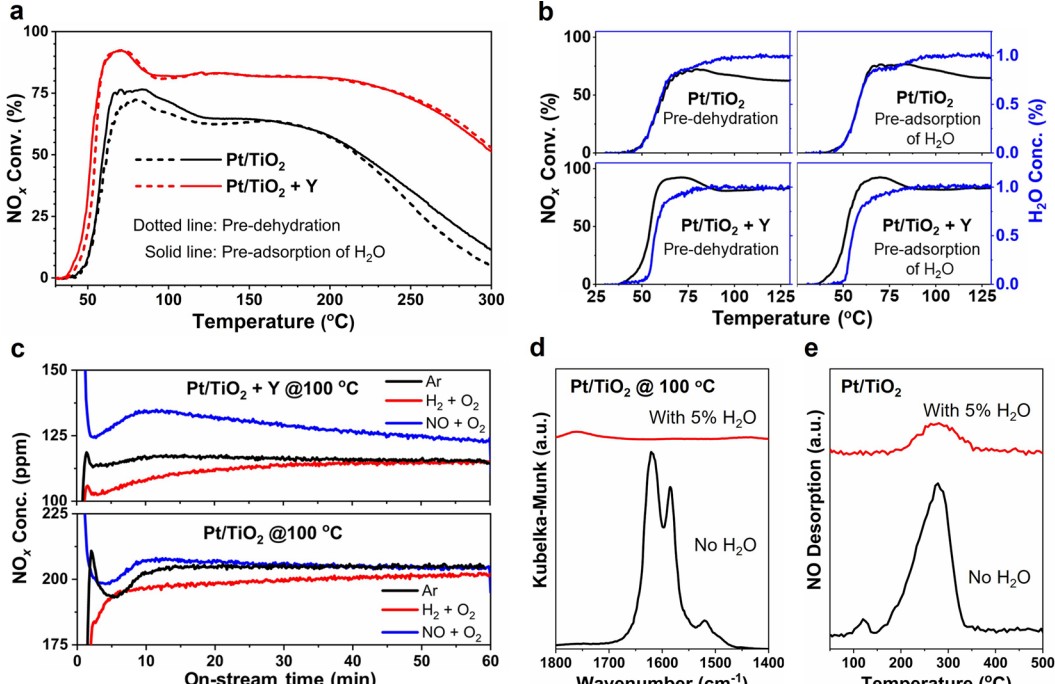

**Fig. 4 | Effects of H₂O on the H₂-SCR performance and NO adsorption property.**
**a** NO$_x$ conversion and (**b**) gas phase $H_2O$ formation during the H₂-SCR reaction over Pt/TiO₂ and Pt/TiO₂ + Y catalysts with pre-dehydration at 300 °C or pre-adsorption of $H_2O$ at 30 °C. Reaction conditions: 26 mg of Pt/TiO₂ catalyst, or a mixture containing 26 mg of Pt/TiO₂ and 26 mg of Y; transient-state light-off testing; 500 ppm NO, 1% H₂, and 10% O₂; WHSV = 461,540 mL·g$_{Pt/TiO2}$$^{-1}$·h$^{-1}$. **c** Time-resolved NO$_x$ concentration

after switching from different flows (Ar; or 1% H₂ + 10% O₂; or 500 ppm NO + 10% O₂) to H₂-SCR flow (500 ppm NO + 1% H₂ + 10% O₂) on Pt/TiO₂ and Pt/TiO₂ + Y catalysts at 100 °C. Testing conditions: 10 mg of Pt/TiO₂, or a mixture containing 10 mg of Pt/TiO₂ and 10 mg of Y; WHSV = 1,200,000 mL·g$_{Pt/TiO2}$$^{-1}$·h$^{-1}$. **d** In situ DRIFTS of NO adsorption at 100 °C, and (**e**) NO-TPD profiles on Pt/TiO₂ catalyst under the NO adsorption conditions with and without 5% $H_2O$.

Pt sites, thereby improving H₂ activation and H₂-SCR performance (Fig. 4a). However, introducing 5% external $H_2O$ into the reaction flow could significantly inhibit the diffusion of NO and H₂ (NO/H₂/$H_2O$ molar ratio = 1/20/100) to the catalyst surface. Despite this inhibition resulting in the decreased activity for both catalysts, the Pt/TiO₂ + Y catalyst still exhibited significantly higher activity compared to the Pt/TiO₂ catalyst (Fig. 1a).

To elucidate the intrinsic promotion effect of Y addition to Pt/TiO₂ on the H₂-SCR performance, systematic density functional theory (DFT) calculations were performed. The Pt (111) surface was selected to represent the Pt active site in Pt/TiO₂ catalyst due to its high thermodynamic stability (Supplementary Fig. 22). As previously confirmed, the Y zeolite in Pt/TiO₂ + Y system possessed high ability to capture the in situ generated $H_2O$, creating the $H_2O$-rich environment around Pt sites. In light of this, a stable $H_2O$/Pt (111) interface was constructed (Supplementary Fig. 23), featuring a hydrogen bonding network with half of $H_2O$ molecules dissociated on Pt with 2/3 monolayer (ML) coverage[41,42]. Such configuration was denoted as the $H_2O$/Pt (111) surface to represent the Pt active site in Pt/TiO₂ + Y system.

As shown in Supplementary Fig. 24, the Pt (111) surface was found more favorable for the NO adsorption with much higher free adsorption energy (−1.77 eV) comparing to that for H₂ adsorption (−0.87 eV) at the low coverage limit. Consequently, the optimal NO coverage on Pt (111) was firstly investigated by calculating the total Gibbs free adsorption energies, which was determined as 7/12 ML at relative low temperatures (T = 320-470 K and $P_{NO}$ = 50 Pa) and in line with previous study[30]. With the highest total Gibbs free adsorption energy, the 7 NO/Pt (111) structure emerged as the most stable configuration (Fig. 5a, Supplementary Fig. 25), which was adopted as the starting point for studying the H₂ activation and reaction mechanism on Pt/TiO₂. On the stable $H_2O$/Pt (111) surface, the presence of a repulsive hydrogen bonding network led to a significant decline in the averaged free NO

adsorption energies. Consequently, a notably reduced NO coverage (1/3 ML) was observed on the $H_2O$/Pt (111) surface (Fig. 5a, Supplementary Fig. 26), in comparison to the NO overage (7/12 ML) on the Pt (111) surface. Considering that the weakly-bonded *NO (T = 373 K, average $G_{ads}$ = -0.5 eV) was highly active and unstable, the $H_2O$/Pt (111) surface without *NO was used as the starting point for studying the H₂ activation and reaction mechanism on Pt/TiO₂ + Y. Accordingly, the H₂ adsorption and activation energies on 7 NO/Pt (111) and $H_2O$/Pt (111) surfaces were calculated, and the results are shown in Fig. 5b. Comparing to the endergonic process of H₂ adsorption (ΔG = 0.30 eV) and high H₂ activation barrier ($G_a$ = 0.75 eV) observed on 7 NO/Pt (111) surface, an exergonic process of H₂ adsorption (ΔG = −0.63 eV) and much lower H₂ activation barrier ($G_a$ = 0.30 eV) was found on $H_2O$/Pt (111) surface. Evidently, the $H_2O$/Pt (111) surface benefited the H₂ adsorption and activation, well aligned with the experimental results showing that Pt/TiO₂ + Y system exhibited superior H₂ activation ability comparing to Pt/TiO₂ (Supplementary Fig. 18).

The theoretical calculations of H₂-SCR reaction mechanism on Pt (111) and $H_2O$/Pt (111) surfaces were conducted to further elucidate the promotion effect of Y zeolite in Pt/TiO₂ + Y system. On Pt (111) surface, as shown in Supplementary Fig. 27 and Supplementary Table 5, the dissociation of *HNOH species into *NH and *OH was found to be the rate-determining step (RDS) for NO reduction with an activation energy ($E_a$) of 1.20 eV. Due to the high NO coverage, the inhibited H₂ activation further hindered the selective reduction of N-containing species to N₂, resulting in the high N₂O formation and low N₂ selectivity. In clear contrast, on $H_2O$/Pt (111) surface (Fig. 6, Supplementary Table 6), the *HNOH species could be readily dissociated into *NH and *OH with a lower activation energy of 0.24 eV (image viii to ix). Once the *NH species was formed, the gas phase NO could facilely couple with it to generate *HNNO, involving a substantial exothermicity of 2.31 eV (image ix to x). Interestingly, rather than releasing N₂O (with

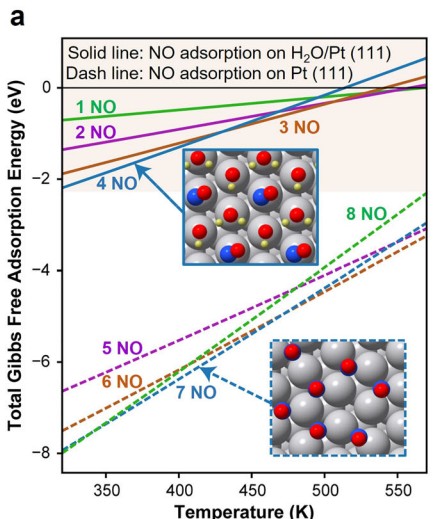

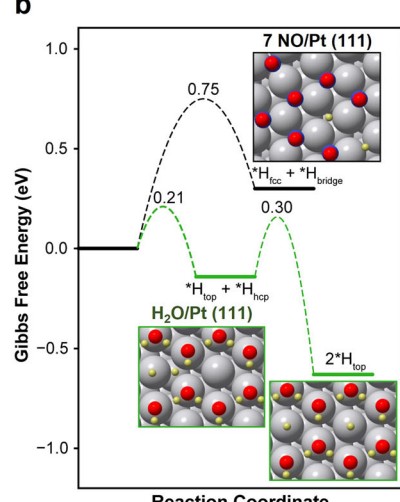

**Fig. 5 | NO adsorption and H₂ activation on Pt (111) and H₂O/Pt (111) surfaces.** **a** Total Gibbs free adsorption energy of NO molecules on Pt (111) and H₂O/Pt (111) surfaces. The reference state is the gas phase NO at 50 Pa. **b** Gibbs free energy diagram of H₂ activation on the 7 NO/Pt (111) surface and the H₂O/Pt (111) surface. The reference state is the gas phase H₂ at 1 atm. Color code: Pt (silver), O (red), N (blue), and H (yellow).

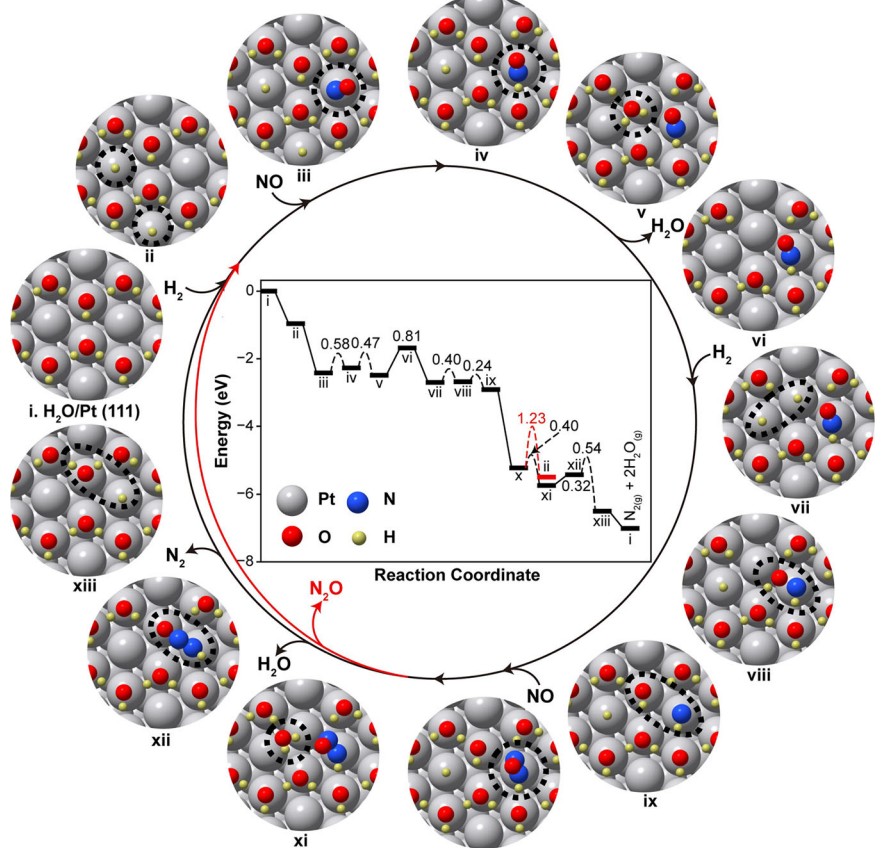

**Fig. 6 | Potential energy diagrams and configurations for the H₂-SCR cycle on the Pt/TiO₂ + Y catalyst.** The reaction was proposed to proceed on the H₂O/Pt (111) surface representing the structure of Pt/TiO₂ + Y catalyst under reaction conditions. The reaction energies and activation energies are indicated in eV in the diagram. Color code: Pt (silver), O (red), N (blue), and H (yellow). Corresponding energies are given in Supplementary Table 6.

$E_a = 1.23$ eV from image x to ii), the *HNNO species remained until an OH vacancy was facilely created in the hydrogen bonding network following the H₂O formation ($E_a = 0.40$ eV from image x to xi) and desorption ($\Delta E = 0.32$ eV from image xi to xii). Subsequently, the *HNNO species was activated and dissociated, selectively producing N₂ with a barrier of 0.54 eV (image xii to xiii). Therefore, the RDS for NO

reduction on H₂O/Pt (111) surface included both the creation of OH vacancy in the hydrogen bonding network and N₂ formation, with an overall activation energy of 0.86 eV (from image xi to xiii). Such activation energy for the RDS of H₂-SCR reaction on H₂O/Pt (111) surface was much lower than that on Pt (111) surface (1.20 eV). These simulation results well explained the significant promotion effect of Y zeolite

in Pt/TiO$_2$ + Y system for H$_2$-SCR in terms of both enhanced NO removal efficiency and elevated N$_2$ selectivity.

## Discussion

A facile, universal and sustainable strategy of physically mixing Pt/oxide catalysts with zeolites has been successfully developed to improve the H$_2$-SCR performance of Pt-based catalysts for low-temperature NO$_x$ removal. The Pt/TiO$_2$ + Y system exhibited superior H$_2$-SCR performance consistently in terms of NO$_x$ conversion and N$_2$ selectivity, both before and after hydrothermal aging, as well as across various testing conditions. This catalyst system shows immense potential in H$_2$-SCR applications, particularly for H$_2$-ICE emission control. It was discovered that the incorporation of Y zeolite effectively promoted H$_2$O adsorption and the formation of H$_2$O-rich environment surrounding Pt active sites in Pt/TiO$_2$ + Y system. This consequently led to the reduction in excessive NO coverage and the improvement in H$_2$ activation, yielding substantial advantages for boosting both H$_2$-SCR efficiency and N$_2$ selectivity. In contrast to modifying the active sites through chemical methods, this study underscores the crucial importance of fine tuning the surrounding environment of active sites through an easy, sustainable physical mixing approach to achieve proficient heterogeneous catalysis.

## Methods

### Catalyst preparation

The Pt/oxide catalysts used in this study, including Pt/TiO$_2$, Pt/Al$_2$O$_3$, and Pt/SiO$_2$, were prepared using incipient wetness impregnation (IWI) method. A solution of colloidal Pt (2-6 nm) with 1 wt% Pt was added dropwise onto commercial anatase TiO$_2$ (surface area = 90 m$^2$/g), γ-Al$_2$O$_3$ (surface area = 150 m$^2$/g), or SiO$_2$ (surface area = 180 m$^2$/g) under stirring, followed by drying at 120 °C for 1 h. After calcination in air at 550 °C for 2 h with the temperature ramp of 5 °C/min, the catalysts were obtained and denoted as Pt/TiO$_2$, Pt/Al$_2$O$_3$, and Pt/SiO$_2$, respectively. As a reference, Pt/Y catalyst was also prepared by the same method using H-Y zeolite (SiO$_2$/Al$_2$O$_3$ molar ratio = 30) as support.

For physical mixing with Pt/oxide catalysts, commercial zeolites including H-Y (SiO$_2$/Al$_2$O$_3$ molar ratio = 30), H-ZSM-5 (SiO$_2$/Al$_2$O$_3$ molar ratio = 30), H-chabazite (CHA, SiO$_2$/Al$_2$O$_3$ molar ratio = 29), H-ferrierite (FER, SiO$_2$/Al$_2$O$_3$ molar ratio = 30), and H-Beta (SiO$_2$/Al$_2$O$_3$ molar ratio = 25) were used. TiO$_2$ or H-Y was also used to dilute Pt/TiO$_2$ or Pt/Y, respectively, for comparison. The content of additional zeolite/oxide was typically controlled at 50 wt% in the physically mixed samples, except for the Pt/TiO$_2$ + Y system with different Y contents of 33, 50, and 67 wt%. These mixed samples were denoted as Pt/oxide + zeolite or oxide. To simulate the catalyst throughout its operational lifespan in heavy-duty vehicles powered by diesel or hydrogen fuel, accelerated aging treatment under hydrothermal conditions of 550–650 °C for 50–100 h should be conducted. In this study, an aging treatment under 10% H$_2$O and 10% O$_2$ at 650 °C for 50 h was performed, and the resulting catalysts were labeled with "-Aged".

### Catalyst characterizations

X-ray diffraction (XRD) measurement was performed on a PANalytical Empyrean diffractometer using a Cu Kα radiation source (λ = 0.15406 nm). The measurement covered the 5° to 80° range with a scan mode of 6 °/min and a scan step of 0.067°.

N$_2$ physisorption was used to determine the surface area, pore volume, and pore size distribution, which was performed on a Quantachrome Autosorb-iQ instrument at liquid nitrogen temperature (77 K). Prior to measurement, all samples were degassed at 300 °C for 2 h under vacuum. The N$_2$ adsorption-desorption isotherm was measured with 40 adsorption and 40 desorption points for Y and Pt/TiO$_2$ + Y samples, and with 20 adsorption and 20 desorption points for Pt/TiO$_2$ using the pressure intervals of 0 < P/P$_0$ < 1. The surface area

was calculated using the Brunauer–Emmett–Teller (BET) method based on the adsorption points in the relative pressure range between 0.05 and 0.3. The Horvath-Kawazoe (HK) method and non-local density functional theory (DFT) method were used to determine the pore volume and pore size distribution.

Transmission electron microscopy (TEM) and energy dispersive X-ray spectroscopy (EDS) mapping images were collected on a field emission FEI Tecnai F-30 with HAADF/ADF/BF STEM and EDS detectors operated at 200 kV.

The CO chemisorption measurement was performed on a Quantachrome Autosorb-iQ instrument. Before each measurement, the sample was first exposed to flowing He from room temperature to 150 °C at the ramp rate of 5 °C/min, and then held at 150 °C for 10 min. Next, the system was purged with 10% H$_2$/Ar, and the temperature was ramped to 400 °C at the rate of 5 °C/min and kept for 30 min. It is important to note that a certain degree of Pt sintering might occur during this reduction treatment, potentially resulting in a lower-estimated Pt dispersion value. The system was then switched back to He, while maintaining the temperature at 400 °C for 30 min. The final step involved cooling the system down to 35 °C in He at the rate of 20 °C/min, holding at 35 °C for 30 min, and injecting multiple CO pulses (5% CO/He) using thermal conductivity detector (TCD) to monitor the gas phase CO.

The X-ray absorption near-edge structure (XANES) and extended X-ray absorption fine structure (EXAFS) of Pt L$_3$-edge were measured at room temperature in fluorescent mode at beamline 7-BM QAS of the National Synchrotron Light Source II (NSLS-II), Brookhaven National Laboratory. Pt foil was measured during data collection for energy calibration and drift correction of the monochromator. Data analysis was conducted using Athena and Artemis from the Demeter software package. The processed EXAFS, χ(k), was weighted by $k^2$ to amplify the high-k oscillations. For Fourier-transformed (FT) spectra, the k range between 3.0 and 12.0 Å was used, and the curve fitting was performed using the Artemis software.

X-ray photoelectron spectroscopy (XPS) was measured on a Thermo Scientific ESCALAB 250Xi photoelectron spectrometer using Al K-α (hν = 1486.68 eV) as the X-ray source in ultrahigh vacuum condition (10$^{-7}$ Pa). The binding energy (BE) of Pt 4d spectra was corrected using the C 1s signal at 284.6 eV as reference.

H$_2$ temperature-programmed reduction (H$_2$-TPR) was performed on the Quantachrome Autosorb-iQ instrument. Prior to testing, the samples were pretreated in a flow of 5% O$_2$/He at 300 °C for 1 h. After cooling down to 40 °C, a flow of 10% H$_2$/Ar was used, and the temperature was raised linearly from 40 to 700 °C at the ramp rate of 10 °C/min. The H$_2$ consumption was monitored on-line using TCD.

In situ DRIFTS experiments were performed on a Nicolet iS50 FTIR spectrometer equipped with a liquid nitrogen-cooled mercury-cadmium-telluride (MCT) detector and an in situ IR cell with ZnSe windows (DiffusIR, PIKE Technologies). Prior to measurements, all samples were pretreated in Ar flow at 300 °C for 1 h. The background spectra at different temperatures (e.g., 25, 100, 125, 150, 175, 200, 225, 250, 275, and 300 °C) were collected in Ar flow using 100 scans with a resolution of 4 cm$^{-1}$. For in situ DRIFTS of CO adsorption at 25 °C, 1% CO/Ar was introduced into the IR cell and kept for 30 min. Then, the samples were purged by Ar for 30 min to remove the weakly adsorbed CO, followed by spectra collection. For in situ DRIFTS of NO adsorption/desorption, the feed stream of 1000 ppm NO, 10% O$_2$, and 5% H$_2$O (when used) in Ar was introduced into the cell with a flow rate of 50 mL/min and kept for 60 min to achieve the saturated NO adsorption at 100 °C. The NO flow was then discontinued while Ar (50 mL/min) was kept flowing for 30 min to remove the gaseous and weakly adsorbed NO. Afterwards, the desorption experiments were carried out in Ar flow with the temperature elevated from 100 to 300 °C with an interval of 25 °C, and the spectra were collected under steady state accordingly. For in situ DRIFTS of H$_2$O adsorption, a feed stream of 5%

$H_2O$ in Ar was introduced into the cell at a flow rate of 50 mL/min and kept for 60 min, achieving saturated $H_2O$ adsorption at 120 °C. Then, the sample was purged with Ar for 30 min at 120 °C to remove the weakly adsorbed $H_2O$, and a background spectrum was collected. The sample was finally treated in Ar flow at 500 °C for 2 h and cooled down to 120 °C, followed by the spectrum collection.

NO temperature-programmed desorption (NO-TPD) and $H_2O$ temperature-programmed desorption ($H_2O$-TPD) were conducted on a continuous flow fixed-bed system. A quartz tubular microreactor with an internal diameter of 4.0 mm was used, and a Hidden Analytical mass spectrometer (MS) was employed as detector. Typically, a feed stream of 1000 ppm NO, 10% $O_2$, and 5% $H_2O$ (when used) in Ar was introduced into the reactor at a flow rate of 40 mL/min and kept for 60 min, achieving saturated NO adsorption at 50 °C. Afterwards, the sample was purged with Ar (40 mL/min) for 120 min at 50 °C to remove the weakly adsorbed molecules. The temperature was then elevated linearly from 50 to 600 °C at a ramp rate of 10 °C/min. For $H_2O$-TPD, a feed stream of 5% $H_2O$ in Ar was introduced into the reactor at a flow rate of 40 mL/min and kept for 60 min, achieving saturated $H_2O$ adsorption at 50 °C. The sample was then purged with Ar (40 mL/min) for 120 min at 50 °C to remove the weakly adsorbed $H_2O$. Subsequently, the temperature was elevated linearly from 50 to 600 °C at a ramp rate of 10 °C/min. The NO or $H_2O$ desorption was monitored on-line using $m/z$ of 30 or 18, respectively.

### Catalytic performance evaluation
The catalytic activity evaluation for the $H_2$-SCR of $NO_x$ over all catalysts was conducted using a continuous flow fixed-bed quartz tubular microreactor with an internal diameter of 4.0 mm. In each test, the catalyst or physical mixture containing 26 mg of Pt/oxide catalyst (40–60 mesh) was diluted with 0.25 g of inert SiC (40-60 mesh) to minimize the effect of hot spots. The reaction atmosphere comprised of 500 ppm NO, 1% $H_2$, 10% $O_2$, 5% $CO_2$ (when used) and 5% $H_2O$ (when used), using Ar as balance. The total flow rate was controlled at 200 mL/min, resulting in a weight hourly space velocity (WHSV) of 461,540 mL·$g_{Pt/oxide}^{-1}$·$h^{-1}$. During the steady-state testing, the catalyst was held at each temperature for a duration of 30 min. Reactants and products were analyzed online by a MultiGas 2030 CEM-Cert FTIR spectrometer. The reactant conversion was defined as ($c_{inlet}$ − $c_{outlet}$)/$c_{inlet}$ × 100%, where $c_{inlet}$ and $c_{outlet}$ were the inlet and outlet $NO_x$ concentration in the feed stream, respectively. The $N_2$ selectivity was defined as $([NO]_{inlet} + [NO_2]_{inlet} − [NO]_{outlet} − [NO_2]_{outlet} − 2 × [N_2O]_{outlet})/([NO]_{inlet} + [NO_2]_{inlet} − [NO]_{outlet} − [NO_2]_{outlet}) × 100\%$. Under the $H_2$-SCR testing conditions with 1% $H_2$ and 10% $O_2$, NO could be either selectively reduced by $H_2$ to form $N_2/N_2O$ or oxidized by $O_2$ to form $NO_2$. The NO selective conversion attributed to the NO reduction by $H_2$ (NO + $H_2$) under the $H_2$-SCR condition was defined as $([NO]_{inlet} + [NO_2]_{inlet} − [NO]_{outlet} − [NO_2]_{outlet})/([NO]_{inlet} − [NO]_{outlet}) × 100\%$, and the NO conversion attributed to the NO oxidation by $O_2$ (NO + $O_2$) under the $H_2$-SCR condition was defined as $([NO_2]_{outlet} − [NO_2]_{inlet})/([NO]_{inlet} − [NO]_{outlet}) × 100\%$. To avoid the significant heat or mass transfer limitation, the kinetics study was performed at 100 °C under the WHSV of 2,400,000 mL·$g_{Pt/TiO_2}^{-1}$·$h^{-1}$ to determine the NO, $H_2$, and $O_2$ reaction orders on Pt/$TiO_2$ and Pt/$TiO_2$ + Y catalysts. The catalytic performance evaluations for separate NO oxidation, $H_2$-SCR in the presence of $NO_2$, $H_2$-SCR in the presence of $NH_3$, separate $H_2$ oxidation, as well as the $H_2$-SCR reaction on the catalysts with pre-dehydration and pre-adsorption of $H_2O$, were also conducted. The detailed information can be found in Supplementary Text 1.

### DFT calculations
Periodic non-spin-polarized DFT calculations were performed using the Vienna Ab-initio Simulation Package (VASP) and the Perdew-Burke-Ernzerhof functional within generalized gradient approximation (GGA). The valence electrons were described by projector augmented wave pseudopotentials with an energy cutoff of 400 eV for all the calculations. The Methfessel-Paxton smearing scheme was used with a width of 0.15 eV and the precision was set to "accurate". The convergence criteria for energies and forces in structure optimizations were set as $10^{-5}$ eV and 0.02 eV Å$^{-1}$, respectively. The van der Waals (vdW) interactions were included via using Grimme's DFT-D3 method. The Brillouin zone for periodic slab calculations was sampled on Γ-centered Monkhorst-Pack type 2 × 3 × 1 k-point grid. Transition states of surface reactions were searched by the nudged elastic band (NEB) together with the dimer method. Further vibrational analysis was adopted to confirm the transition states. Only one imaginary frequency mode along the reaction trajectory represented the true saddle point.

The reaction energy ($\Delta E$) of each elementary step was computed by the difference between the DFT energy of the final state ($E_{FS}$) and that of the corresponding initial state ($E_{IS}$), with $\Delta E = E_{FS} − E_{IS}$. Similarly, the activation energy was calculated using the equation, $E_a = E_{TS} − E_{IS}$, where $E_{TS}$ was the DFT energy of corresponding transition state (TS). H binding energy ($E_b(H)$) was computed by the equation, $E_b(H) = E_{H/support} − E_{support} − 0.5E_{H2}$, where $E_{H/support}$, $E_{support}$ and $E_{H2}$ were the DFT energies of support with the *H adsorbate, the support, and gas phase $H_2$, respectively. Gibbs free energy of each species was calculated by

$$G = E + E_{ZPE} + C_p T − TS \qquad (1)$$

in which $G$ was the Gibbs free energy, and $E$, $E_{ZPE}$, $C_p$ and $S$ were the DFT energy, zero point energy, heat capacity and entropy of each gas-phase species or surface intermediates, respectively. The $E_{ZPE}$, $C_p$, and $S$ were calculated within the harmonic approximation. The Atomic Simulation Environment (ASE) package was employed to calculate the Gibbs free energy of gas and adsorbed species at certain temperatures and pressures.

The Gibbs free formation energies of adsorbates on corresponding surface were calculated via the following equation:

$$G_f(N_xO_yH_z/surface) = G(N_xO_yH_z/surface) − G(surface) − xG(NO)$$
$$−(y − x) × G(H_2O) − (z/2 − y + x) × G(H_2) \qquad (2)$$

in which $G(N_xO_yH_z/surface)$, $G(surface)$, $G(NO)$, $G(H_2O)$, and $G(H_2)$ were the Gibbs free energies of the surface with adsorbates, the clean surface, and gas phase NO, $H_2O$, and $H_2$ under relevant temperatures and pressures, respectively. The partial pressures of gas phase NO, $H_2$, and $H_2O$ were set as 50, 1000, and 5000 Pa, which were within the range of experimental operation conditions.

### Reporting summary
Further information on research design is available in the Nature Portfolio Reporting Summary linked to this article.

## Data availability
Source data are provided with this paper.

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

## Acknowledgements

This work was supported by a research fund from BASF Environmental Catalyst and Metal Solutions and the Startup Fund (F.L.) from the University of California, Riverside (UCR). S.X., D.K. and X.Z. thank the support from the Preeminent Postdoctoral Program (P3) at the University of Central Florida (UCF). L.L. and H.X. thank the support from NSF CDS&E program (CBET-2245402). F.L. sincerely thanks Mr. Franck Thibaut and Ms. Corinne Lehaut from Tronox Inc., Dr. Marcos Schöneborn from Sasol, and Dr. Chris Bauer from Evonik for providing raw materials in catalyst synthesis. F.L. and S.X. thank Dr. Tangyuan Li and Prof. Liangbing Hu from the University of Maryland for their assistance with N$_2$

physisorption testing. This research used beamline 7-BM (QAS) of the National Synchrotron Light Source II, a U.S. Department of Energy (DOE) Office of Science User Facility operated for the DOE Office of Science by Brookhaven National Laboratory under Contract No. DE-SC0012704. H.X. acknowledges the computational resource provided by the advanced research computing at Virginia Polytechnic Institute and State University.

## Author contributions

F.L. and Y.L. conceived the idea and directed the project. S.X. designed the experiments, performed the experiments, and analyzed the data. K.Y., D.K. and X.Z. assisted with the catalyst testing and characterization. L.M. and S.E. conducted XAS measurements. L.L. and H.X. performed DFT calculations and analysis. S.X. and L.L. wrote the manuscript. F.L. and H.X. mentored the manuscript writing and revision. All authors discussed the results and commented on the manuscript.

## Competing interests

The authors declare no competing interests.
