## [Peer Review File · Nature Communications]

Zeolite-Promoted Platinum Catalyst for Efficient Reduction of Nitrogen Oxides with HydrogenREVIEWER COMMENTS

Reviewer #1 (Remarks to the Author):

In this article, authors investigated an interesting effect by trapped water in Y zeolite for H₂-SCR on Pt/TiO₂. By simply mixing Pt/TiO₂ and Y zeolite, a better H₂-SCR and N₂ selectivity can be achieved. In addition to this novel finding, authors thoroughly studied the kinetics of the reactions and further confirmed the observed effect and the mechanism. The paper was well written and is recommended to be accepted.

Just one minor suggestion, the authors investigated other zeolite systems and briefly mentioned in the paper. Since the paper focuses on Y zeolite, it might not be necessary to include others. The paper has a significant amount of figures and data even without these data from other zeolites.

Reviewer #2 (Remarks to the Author):

The manuscript "Miracle Created Through Physical Mixing: Zeolite-Promoted Platinum Catalyst for Efficient Reduction of Nitrogen Oxides with Hydrogen" by Professor Fudong Liu et al. report that catalytic performance of Pt/TiO₂ with commercial Y zeolite for H₂-SCR. It is preferable that this paper be revised on the following points before acceptance.

- Abstract

"The incorporation of Y zeolite effectively facilitated the capture of in-situ generated water, fostering a water-rich environment surrounding the Pt active sites within Pt/TiO₂ + Y catalyst."

Water (steam) usually deactivates catalytic activity, but the opposite trend is reported. The reason for this should be clarified.

- Catalyst preparation

"Hydrothermal aging was performed for the catalysts under 10% H₂O and 10% O₂ at 650 °C for 50 h"

The authors should discuss the reasons for this hydrothermal aging conditions. Since 500 ppm NO is thought to be generated by thermal NO_x, the exhaust temperature is thought to be much higher.

- Results

"Fig. 1"

NO conversion and N₂ selectivity have not reached more than 99%. Clarifications are required on the possible byproduction of NO₂ and NH₃ as well.

"These results evidently suggest the critical role of establishing the close contact between Pt/TiO₂ and Y zeolite in enhancing the overall H₂-SCR performance."

Because the close contact between Pt/TiO₂ and Y zeolite is important, wet mixing rather than physical mixing would be expected to improve catalytic performance.

“Fig. 2”

Despite the catalysts being prepared by the wet impregnation method and the high surface area for support materials, it is considered that Pt dispersions are low.

The reviewer hopes that the authors reconsider.

Reviewer #3 (Remarks to the Author):

Shaohua Xie et al.

Miracle Created Through Physical Mixing: Zeolite-Promoted Platinum Catalyst for Efficient Reduction of Nitrogen Oxides with Hydrogen

This manuscript deals with the strategy of using a physical mixing of Pt/TiO₂ with commercial Y zeolite to achieve a notable enhancement of NO_x conversion at low temperature and N₂ selectivity in H₂-ICE exhaust gas streams by H₂-SCR. This is a facile, universal and sustainable strategy, which shows substantial potential for NO_x removal in H₂-SCR applications, but no clear advantages are demonstrated vs. specific catalyst systems including the role of the active metal phase and water adsorption component on a single sample, to tune the local reaction environment around the active sites. Also, deposition of such physical mixture on structured supports (monoliths or similar, usual in catalytic converters) is not obvious to maintain efficiency. In general terms, I agree with initial feelings of Editor that neither the strategy nor the formulation, nor the findings represent a sufficiently striking advance to justify publication in Nature Communication, even after reading the authors' rebuttal letter. Nevertheless, the work is well-structured, experiments well-designed and results and discussion are conducting to relevant results, thus I think the manuscript merits publication in a dedicated catalysis magazine such as Applied Catalysis, Catalysis Today, Topics in Catalysis, ChemCatChem, ...

Significant research has been conducted in recent years to investigate the correlation between the properties and structure of H₂-SCR. Depending on the chemical promoter and support composition, Pd-based catalysts in stationary NO_x control applications may reach N₂-selectivity and NO_x conversion of 80-95% (Pt-based catalysts exhibited lower values). Different Pd-based catalyst compositions yield over 85% NO_x conversion in mobile applications. Authors can find a systematic review on “Innovative catalysts for the selective catalytic reduction of NO_x with H₂” (Farhan et al. *Fuel* 355 (2024) 129364), covering different types of catalysts, the significant impact of catalyst chemical components, including the active component (comprising noble and non-noble metals, and bimetallic compositions), supports (including zeolites and titania). The study also provides a comprehensive discussion on the effect of exhaust gas composition, including O₂, CO, CO₂, H₂O, ...

Zeolites offering good thermal stability have been used for NH₃-SCR, so that several groups suggested zeolitic and zeolite-related systems also in the field of H₂-SCR. Borchers et al (*Top. Catal.* doi: 10.1007/s11244-022-01723-1) include TiO₂ and Y zeolite in a single Pd-based catalyst as 1%Pd/20%TiO₂/HY, which outperforms one active and selective benchmark catalyst for H₂-SCR. I am wondering if a similar formulation with Pt (same components than in your sample, but integrated in a single solid) will run under same reaction mechanism

achieving worse/similar/better activity and selectivity. That is to say, the physical mixing Pt/TiO₂ + Y or Pt/TiO₂/Y, which would be preferred? Which will be better to modulate activity, selectivity and durability of the H₂-SCR catalyst? Furthermore, significant attention has been given to the role of noble metals at various sites within the zeolite structure, resulting in a thorough understanding of their effects. Inclusion of this discussion will greatly enhance the scientific insights of the paper.

On the other, the mixing strategy of solids is not new in catalysis for many processes. The mixing of two complementary catalysts is generally made when looking for catalytic synergism (see the concept in Cang and Phillips, *Langmuir* 12 (1996) 2756). On the other hand, the approach given in the current manuscript is used when upgrading industrial catalytic processes without modifying the catalyst itself (Pt/TiO₂ in this case by mixing with commercial Y zeolite), also reported e.g. Fang et al., *Science* 377 (2022) 406.

In the following, I am suggesting some other considerations, in case they could help to authors for improving the manuscript.

1. In SCR metal-based catalysts supported on zeolites, the metal speciation and situation at various positions in the structure of zeolite is a key point to understand mechanism and behaviour. This can be checked in many of the advanced catalysts designed summarized in paper by Farhan et al. (see above). What about a preparation with the Pt/TiO₂ supported on Y zeolite, instead of the physical mixing? If the idea is to surround Pt active sites with acidic sites of zeolite, a better intimate and stable contact should be achieved by deposition or incorporation of Pt/TiO₂ in the zeolite structure.
2. The paper refers to selectivity to N₂ but does not analyse the rest of N-products, neither for Pt-TiO₂ catalyst nor for Pt/TiO₂ + Y mixing strategy. For example at 250 °C where selectivities are 17% and 58%, respectively (Figure 1).
3. EDS mapping images revealed no contact between Pt and acidic sites after physical mixing. Authors should check and confirm if there is any modification in species distribution after reaction has been performed in order to confirm no real atomic contact between them during reaction conditions.
4. Additionally, characterization shown in Table 1 should also be performed on used catalysts in order to discard any possible effect over surface properties.
5. Authors should explain the negative impact of adding increasing amounts of H-Y above 50%. One would expect a saturation effect in the improvement, rather than a NO_x conversion loss.
6. Please, specify if the correct Si/Al ratio for H-Y is 29 or 30, both values appear along the manuscript.
7. Please, could you suggest the final De-NO_x mechanism derived from your testing experiments? How is affecting the physical mixing to the mechanism with single Pt/TiO₂ catalyst? Is it more related to a change in composition of NO_x stream accessing to Pt/TiO₂ active sites, not related to the surface of the catalyst?

Reviewer #4 (Remarks to the Author):

This manuscript deals with the strategy of using a physical mixing of Pt/TiO₂ with commercial Y zeolite to achieve a notable enhancement of NO_x conversion at low temperature and N₂ selectivity in H₂-ICE exhaust gas streams by H₂-SCR. This is a facile,

universal and sustainable strategy, which shows substantial potential for NO_x removal in H₂-SCR applications, but no clear advantages are demonstrated vs. specific catalyst systems including the role of the active metal phase and water adsorption component on a single sample, to tune the local reaction environment around the active sites. Also, deposition of such physical mixture on structured supports (monoliths or similar, usual in catalytic converters) is not obvious to maintain efficiency. In general terms, I agree with initial feelings of Editor that neither the strategy nor the formulation, nor the findings represent a sufficiently striking advance to justify publication in Nature Communication, even after reading the authors' rebuttal letter. Nevertheless, the work is well-structured, experiments well-designed and results and discussion are conducting to relevant results, thus I think the manuscript merits publication in a dedicated catalysis magazine such as Applied Catalysis, Catalysis Today, Topics in Catalysis, ChemCatChem, ...

Significant research has been conducted in recent years to investigate the correlation between the properties and structure of H₂-SCR. Depending on the chemical promoter and support composition, Pd-based catalysts in stationary NO_x control applications may reach N₂-selectivity and NO_x conversion of 80-95% (Pt-based catalysts exhibited lower values). Different Pd-based catalyst compositions yield over 85% NO_x conversion in mobile applications. Authors can find a systematic review on "Innovative catalysts for the selective catalytic reduction of NO_x with H₂" (Farhan et al. Fuel 355 (2024) 129364), covering different types of catalysts, the significant impact of catalyst chemical components, including the active component (comprising noble and non-noble metals, and bimetallic compositions), supports (including zeolites and titania). The study also provides a comprehensive discussion on the effect of exhaust gas composition, including O₂, CO, CO₂, H₂O, ...

Zeolites offering good thermal stability have been used for NH₃-SCR, so that several groups suggested zeolitic and zeolite-related systems also in the field of H₂-SCR. Borchers et al (Top. Catal. doi: 10.1007/s11244-022-01723-1) include TiO₂ and Y zeolite in a single Pd-based catalyst as 1%Pd/20%TiO₂/HY, which outperforms one active and selective benchmark catalyst for H₂-SCR. I am wondering if a similar formulation with Pt (same components than in your sample, but integrated in a single solid) will run under same reaction mechanism achieving worse/similar/better activity and selectivity. That is to say, the physical mixing Pt/TiO₂ + Y or Pt/TiO₂/Y, which would be preferred? Which will be better to modulate activity, selectivity and durability of the H₂-SCR catalyst? Furthermore, significant attention has been given to the role of noble metals at various sites within the zeolite structure, resulting in a thorough understanding of their effects. Inclusion of this discussion will greatly enhance the scientific insights of the paper.

On the other, the mixing strategy of solids is not new in catalysis for many processes. The mixing of two complementary catalysts is generally made when looking for catalytic synergism (see the concept in Cang and Phillips, Langmuir 12 (1996) 2756). On the other hand, the approach given in the current manuscript is used when upgrading industrial catalytic processes without modifying the catalyst itself (Pt/TiO₂ in this case by mixing with commercial Y zeolite), also reported e.g. Fang et al., Science 377 (2022) 406.

In the following, I am suggesting some other considerations, in case they could help to authors for improving the manuscript.

1. In SCR metal-based catalysts supported on zeolites, the metal speciation and situation at various positions in the structure of zeolite is a key point to understand mechanism and behaviour. This can be checked in many of the advanced catalysts designed summarized in

paper by Farhan et al. (see above). What about a preparation with the Pt/TiO₂ supported on Y zeolite, instead of the physical mixing? If the idea is to surround Pt active sites with acidic sites of zeolite, a better intimate and stable contact should be achieved by deposition or incorporation of Pt/TiO₂ in the zeolite structure.

2. The paper refers to selectivity to N₂ but does not analyse the rest of N-products, neither for Pt-TiO₂ catalyst nor for Pt/TiO₂ + Y mixing strategy. For example at 250 °C selectivities are 17% and 58%, respectively (Figure 1).

3. EDS mapping images revealed no contact between Pt and acidic sites after physical mixing. Authors should check and confirm if there is any modification in species distribution after reaction has been performed in order to confirm no real atomic contact between them during reaction conditions.

4. Additionally, characterization shown in Table 1 should also be performed on used catalysts in order to discard any possible effect over surface properties.

5. Authors should explain the negative impact of adding increasing amounts of H-Y above 50%. One would expect a saturation effect in the improvement, rather than a NO_x conversion loss.

6. Please, specify if the correct Si/Al ratio for H-Y is 29 or 30, both values appear along the manuscript.

7. Please, could you suggest the final De-NO_x mechanism derived from your testing experiments? How is affecting the physical mixing to the mechanism with single Pt/TiO₂ catalyst? Is it more related to a change in composition of NO_x stream accessing to Pt/TiO₂ active sites, not related to the surface of the catalyst?

Notes:

According to the comments and suggestions from the reviewers, the Main Manuscript and the Supplementary Information have been carefully revised. All the changes are indicated using **red fonts** for the reviewers' convenience.

Point-to-Point Response to the Reviewers' Comments:

Reviewer #1 (Remarks to the Author):

In this article, authors investigated an interesting effect by trapped water in Y zeolite for H₂-SCR on Pt/TiO₂. By simply mixing Pt/TiO₂ and Y zeolite, a better H₂-SCR and N₂ selectivity can be achieved. In addition to this novel finding, authors thoroughly studied the kinetics of the reactions and further confirmed the observed effect and the mechanism. The paper was well written and is recommended to be accepted. Just one minor suggestion, the authors investigated other zeolite systems and briefly mentioned in the paper. Since the paper focuses on Y zeolite, it might not be necessary to include others. The paper has a significant amount of figures and data even without these data from other zeolites.

Response: Thank you very much for your kind suggestion. We acknowledge that our manuscript contains a substantial number of figures and data. Considering our aim to present a more comprehensive overview of our concept, we believe that including the data on other zeolites in addition to Y zeolite can be beneficial. Hope you can kindly understand us. We appreciate your recommendation of publishing our manuscript on *Nature Communications*.

Reviewer #2 (Remarks to the Author):

The manuscript "Miracle Created Through Physical Mixing: Zeolite-Promoted Platinum Catalyst for Efficient Reduction of Nitrogen Oxides with Hydrogen" by Professor Fudong Liu et al. report that catalytic performance of Pt/TiO₂ with commercial Y zeolite for H₂-SCR. It is preferable that this paper be revised on the following points before acceptance.

(1) - Abstract

"The incorporation of Y zeolite effectively facilitated the capture of in-situ generated water, fostering a water-rich environment surrounding the Pt active sites within Pt/TiO₂ + Y catalyst." Water (steam) usually deactivates catalytic activity, but the opposite trend is reported. The reason for this should be clarified.

Response: Thank you for your valuable suggestion. Yes, indeed, the presence of a high concentration of water in the reaction flow typically affects catalytic activity negatively in two ways: it inhibits the diffusion of reactants to the metal surface and reduces reactant adsorption, and/or blocks metal active sites due to strong water adsorption. However, the impact on supported Pt catalysts depends on the specific reaction. For instance, in reactions such as CO oxidation (*ACS Catal.* 2017, 7, 1, 887-891; *J. Catal.* 2020, 382, 192-203), oleic acid conversion (*Catal. Commun.* 2017, 98, 26-29), and NO reduction (*Appl. Catal. B: Environ.* 2010, 97, 236-247), a promotional effect of water was observed. In this work, water is a product of the H₂-SCR reaction. Under the testing conditions without external water addition, we observed that the incorporation of Y zeolite into the Pt/TiO₂ catalyst created a water-rich environment around the Pt active sites by capturing the *in-situ* generated water (Fig. 4b). This environment resulted in weakened NO adsorption and

improved H₂ activation with water adsorbing on the Pt sites, as verified by both experimental and theoretical studies. Consequently, the Pt/TiO₂ + Y catalyst exhibited significantly elevated H₂-SCR activity and N₂ selectivity (Fig. 4a). When introducing 5% water into the reaction flow, the diffusion of NO and H₂, which were present in much lower concentrations (NO/H₂/H₂O molar ratio = 1/20/100), could be significantly inhibited. This inhibition led to a decrease in activity for both the Pt/TiO₂ and Pt/TiO₂ + Y catalysts. It is noteworthy that the Pt/TiO₂ + Y catalyst consistently outperformed the Pt/TiO₂ catalyst under various testing conditions (Figs. 1a and 4a), demonstrating the significant promotional effect of Y on the H₂-SCR activity.

Modification: Top/Page 11: The physical mixing of Pt/TiO₂ with Y could further promote the adsorption of *in situ* generated H₂O (Fig. 3e and f), creating a H₂O-rich environment around the Pt sites and facilitating the formation of a H₂O-covered Pt surface. This surface could reduce the NO coverage on Pt sites, thereby improving H₂ activation and H₂-SCR performance (Fig. 4a). However, introducing 5% external H₂O into the reaction flow could significantly inhibit the diffusion of NO and H₂ (NO/H₂/H₂O molar ratio = 1/20/100) to the catalyst surface. Despite this inhibition resulting in the decreased activity for both catalysts, the Pt/TiO₂ + Y catalyst still exhibited significantly higher activity compared to the Pt/TiO₂ catalyst (Fig. 1a).

2, - Catalyst preparation

“Hydrothermal aging was performed for the catalysts under 10% H₂O and 10% O₂ at 650 °C for 50 h”

The authors should discuss the reasons for this hydrothermal aging conditions. Since 500 ppm NO is thought to be generated by thermal NO_x, the exhaust temperature is thought to be much higher.

Response: Thank you for your kind comments. During the combustion process, the temperature within the combustion chambers of internal combustion engines typically ranges from about 900 °C to 2500 °C, which results in the formation of NO_x, as NO_x formation typically begins to occur significantly above 1600 °C. However, the catalyst for exhaust treatment is placed downstream the combustion chambers, where the temperatures are typically below 500 °C (*Energies* 2021, 14, 8166), especially for heavy-duty vehicle applications. Hydrothermal treatment is usually used to accelerate the catalyst aging process to simulate the long-term operation conditions. Since H₂-ICEs show great potential to replace heavy-duty diesel engines, the typical hydrothermal aging conditions for heavy-duty diesel applications were used in this work. Specifically, the required accelerated aging conditions from automotive OEMs (Original Equipment Manufacturers) and catalyst suppliers (such as BASF Corporation and Johnson Matthey Inc.) for heavy-duty diesel vehicles are 550-650 °C for 50-100 h. Therefore, the aging treatment condition used in this work (650 °C for 50 h in 10% H₂O and 10% O₂ flow) was reasonable and sufficient for evaluating the newly developed catalysts.

Modification: Middle/Page 15: To simulate the catalyst throughout its operational lifespan in heavy-duty vehicles powered by diesel or hydrogen fuel, accelerated aging treatment under hydrothermal conditions of 550-650 °C for 50-100 h should be conducted. In this study, an aging treatment under 10% H₂O and 10% O₂ at 650 °C for 50 h was performed, and the resulting catalysts were labeled with '-Aged'.

3, - Results

“Fig. 1”

NO conversion and N₂ selectivity have not reached more than 99%. Clarifications are required on the possible byproduct of NO₂ and NH₃ as well.

Response: Thank you for your suggestions. The N-species distributions in different components during H₂-SCR on the Pt/TiO₂ and Pt/TiO₂ + Y catalysts have been added to the revised manuscript. As shown in the figure below (Supplementary Fig. 1), under the high space velocity testing conditions with H₂O and CO₂ (500 ppm NO, 1% H₂, 10% O₂, 5% CO₂, and 5% H₂O; WHSV = 461,540 mL·g_{Pt/TiO₂}⁻¹·h⁻¹), there was always a portion of NO that could not be completely reduced and was further oxidized to form NO₂ at elevated temperatures. Therefore, the NO_x conversion could not achieve 100% and declined as the temperature increased. Additionally, the N₂ selectivity was below 100% due to the formation of N₂O as the only by-product. No NH₃ formation was detected throughout the entire H₂-SCR reaction. Specifically, at 250 °C, Pt/TiO₂ catalyst produced 9.1% N₂O and 1.9% N₂, while Pt/TiO₂ + Y catalyst produced 24.8% N₂O and 34.2% N₂, resulting in the N₂ selectivity of 17% on Pt/TiO₂ and 57% on Pt/TiO₂ + Y. Clearly, the incorporation of Y greatly enhanced the NO_x conversion and N₂ selectivity simultaneously on Pt/TiO₂ catalyst.

Modification: Top/Page 5: In the presence of both H₂ and O₂, NO can either be reduced by H₂ to form N₂/N₂O or be oxidized by O₂ to form NO₂ (Supplementary Fig. 1). Therefore, it is reasonable that the NO_x conversion and N₂ selectivity could hardly achieve 100% under the high space velocity H₂-SCR testing conditions with H₂O and CO₂ (500 ppm NO, 1% H₂, 10% O₂, 5% CO₂, and 5% H₂O; WHSV = 461,540 mL·g_{Pt/TiO₂}⁻¹·h⁻¹).

Supplementary Fig. 1 | N-species distribution. N-species distribution in N₂O, N₂, NO₂, and NO during H₂-SCR on Pt/TiO₂ and Pt/TiO₂ + Y catalysts. Reaction conditions: 26 mg of Pt/TiO₂ catalyst or the mixture containing 26 mg of Pt/TiO₂ catalyst and 26 mg of Y; 500 ppm NO, 1% H₂, 10% O₂, 5% CO₂, and 5% H₂O; WHSV = 461,540 mL·g_{Pt/TiO₂}⁻¹·h⁻¹.

4, “These results evidently suggest the critical role of establishing the close contact between Pt/TiO₂ and Y zeolite in enhancing the overall H₂-SCR performance.”

Because the close contact between Pt/TiO₂ and Y zeolite is important, wet mixing rather than physical mixing would be expected to improve catalytic performance.

Response: Thank you for your constructive suggestions. To verify whether closer contact between Pt, TiO₂, and Y can further improve the H₂-SCR performance, we prepared Pt-TiO₂-Y catalysts

using two additional methods. The first method involved physically mixing Pt/TiO₂ with Y in the presence of water, followed by drying at 300 °C for 2 h. The resulting catalyst is referred as (Pt/TiO₂ + Y)_{H₂O}. The second method involved preparing the TiO₂/Y support by precipitating 50 wt.% TiO₂ from a Ti(OBu)₄ solution in ethanol onto Y. After drying at 120 °C for 1 h and calcination in air at 550 °C for 2 h, colloidal Pt (2-6 nm) with 1 wt.% Pt was added dropwise onto the TiO₂/Y support. Following additional drying at 120 °C for 1 h and calcination in air at 550 °C for 2 h, the catalyst was obtained and referred as Pt/TiO₂/Y. When subjecting these two catalysts to the H₂-SCR reaction, as shown in Supplementary Fig. 3, it was observed that the (Pt/TiO₂ + Y)_{H₂O} catalyst showed slightly higher activity than Pt/TiO₂ + Y catalyst, while the Pt/TiO₂/Y catalyst exhibited lower activity compared to the Pt/TiO₂ + Y catalyst. However, both catalysts demonstrated lower N₂ selectivity than Pt/TiO₂ + Y catalyst. These results indicate that suitable contact (not necessarily the very close contact) between Pt/TiO₂ and Y could benefit both activity and N₂ selectivity. In Pt/TiO₂/Y catalyst, the stronger interaction (probably much closer contact) between Pt, TiO₂, and Y might be present. This interaction could significantly enhance H₂ dissociation, leading to increased H₂-O₂ reaction activity, decreased H₂-NO reaction activity, and reduced N₂ selectivity as well. However, this hypothesis needs to be further investigated in future work.

Modification: Bottom/Page 5: To achieve even closer contact between Pt/TiO₂ and Y, we further physically mixed the Pt/TiO₂ and Y powders in the presence of water, referred as (Pt/TiO₂ + Y)_{H₂O}, and loaded Pt onto a pre-prepared 50% TiO₂/Y support (denoted as Pt/TiO₂/Y). It was observed that the (Pt/TiO₂ + Y)_{H₂O} catalyst showed slightly higher activity (**Supplementary Fig. 3**), and the Pt/TiO₂/Y catalyst exhibited lower activity compared to the Pt/TiO₂ + Y catalyst. However, both catalysts demonstrated lower N₂ selectivity than Pt/TiO₂ + Y catalyst. These results evidently suggest the critical role of establishing an appropriate contact between Pt/TiO₂ and Y zeolite in enhancing the overall H₂-SCR performance.

Supplementary Fig. 3 | The effect of contact between Pt, TiO₂, and Y in Pt-TiO₂-Y system on the H₂-SCR performance. (a) NO_x conversion and (b) N₂ selectivity in H₂-SCR reaction over Pt/TiO₂ + Y, (Pt/TiO₂ + Y)_{H₂O}, and Pt/TiO₂/Y catalysts. Reaction conditions: 52 mg of Pt/TiO₂ + Y and (Pt/TiO₂ + Y)_{H₂O}, or 26 mg of Pt/TiO₂/Y; 500 ppm NO, 1% H₂, 10% O₂, 5% CO₂, and 5% H₂O; WHSV = 461,540 mL·g_{Pt/TiO₂}⁻¹·h⁻¹.

Notes: To verify whether closer contact between Pt, TiO₂, and Y could further improve H₂-SCR performance, two additional Pt-TiO₂-Y catalysts were prepared and tested for the H₂-SCR reaction. The first catalyst, referred as (Pt/TiO₂ + Y)_{H₂O}, was fabricated by physically mixing Pt/TiO₂ with Y in the presence of water, followed by drying at 300 °C for 2 h. The second catalyst, denoted as Pt/TiO₂/Y, was obtained by impregnating colloidal Pt (2-6 nm) with 1 wt.% Pt onto a TiO₂/Y support, followed by drying at 120 °C for 1 h and calcination in air at 550 °C for 2 h. The TiO₂/Y support was prepared by precipitating 50 wt.% TiO₂ from a Ti(OBu)₄ solution in ethanol onto Y, followed by drying at 120 °C for 1 h and calcination in air at 550 °C for 2 h.

5, “Fig. 2”

Despite the catalysts being prepared by the wet impregnation method and the high surface area for support materials, it is considered that Pt dispersions are low.

The reviewer hopes that the authors reconsider.

Response: Thank you very much for your comment. Yes, we agree with your opinion. Due to the usage of colloidal Pt (2-6 nm) as precursor and TiO₂ support with high surface area, the Pt particles with average sizes of 6.0 and 6.2 nm were highly dispersed in both Pt/TiO₂ and Pt/TiO₂ + Y catalysts, as seen from the TEM images. According to semi-sphere model, the Pt dispersion could be calculated as 13% for both catalysts, which was indeed higher than the 8.5-8.9% obtained from the CO pulse titration analysis. Since a pre-reduction at 400 °C in 10% H₂/Ar was conducted on the samples before the CO titration experiment, we believe that there might have been a certain degree of Pt sintering occurring during the reduction treatment, thus leading to the lower Pt dispersion values. We have indicated this possibility in this revised manuscript.

Modification: Top/Page 16: the system was purged with 10% H₂/Ar, and the temperature was ramped to 400 °C at the rate of 5 °C/min and kept for 30 min. It is important to note that a certain degree of Pt sintering might occur during this reduction treatment, potentially resulting in a lower-estimated Pt dispersion value.

Reviewers #3 and #4 (Remarks to the Author):

Shaohua Xie et al. Miracle Created Through Physical Mixing: Zeolite-Promoted Platinum Catalyst for Efficient Reduction of Nitrogen Oxides with Hydrogen

This manuscript deals with the strategy of using a physical mixing of Pt/TiO₂ with commercial Y zeolite to achieve a notable enhancement of NO_x conversion at low temperature and N₂ selectivity in H₂-ICE exhaust gas streams by H₂-SCR. This is a facile, universal and sustainable strategy, which shows substantial potential for NO_x removal in H₂-SCR applications, but no clear advantages are demonstrated vs. specific catalyst systems including the role of the active metal

phase and water adsorption component on a single sample, to tune the local reaction environment around the active sites. Also, deposition of such physical mixture on structured supports (monoliths or similar, usual in catalytic converters) is not obvious to maintain efficiency. In general terms, I agree with initial feelings of Editor that neither the strategy nor the formulation, nor the findings represent a sufficiently striking advance to justify publication in *Nature Communication*, even after reading the authors' rebuttal letter. Nevertheless, the work is well-structured, experiments well-designed and results and discussion are conducting to relevant results, thus I think the manuscript merits publication in a dedicated catalysis magazine such as *Applied Catalysis*, *Catalysis Today*, *Topics in Catalysis*, *ChemCatChem*,...

Response: Thank you for your detailed comments. We believe that the strategy developed in this work indeed offers significant advantages over traditional approaches for catalyst development. Unlike the methods tailored to upgrade the active sites for specific catalyst systems, our approach of physically mixing Pt/oxide catalysts with zeolites provides a simple and universal technique aimed at further enhancing the H₂-SCR performance of well-established and efficient Pt/oxide catalysts. It should be acknowledged that coating powder catalysts onto substrates, such as monoliths, is more complex compared to using powder catalysts alone. The catalytic performance of the coated samples can be influenced by the coating process, binders, slurry pH, and other variables. Optimizing these factors to achieve the best catalytic performance often requires considerable effort, even with an efficient powder catalyst system. As our work mainly focused on developing a new concept catalyst, we did not perform additional evaluations involving coating these catalysts onto monoliths or other substrates in this study. However, our collaborator, BASF Corporation, is interested in this concept and plans to conduct a thorough investigation to explore its potential industrial application. We are confident that this new concept catalyst, even as a physical mixture, holds high promise for practical application in catalytic converters after being coated onto suitable substrates.

We firmly believe that this work, which demonstrates high novelty and significant advancements, deserves to be published in the *Nature Communications*. It presents a new and universal strategy, a catalyst formulation with exceptional catalytic performance, and a profound understanding of this catalyst in H₂-SCR. To our knowledge, the strategy of physically mixing an oxide-supported metal catalyst (e.g., Pt/TiO₂) with a zeolite (e.g., Y) is unprecedented and is reported here for the first time in H₂-SCR application. This strategy is simple, cost-effective, and scalable, demonstrating a universal promotion effect that enhances the H₂-SCR activity of oxide-supported metal catalysts such as Pt/TiO₂, Pt/Al₂O₃, and Pt/SiO₂, making it highly promising for industrial applications. The representative catalyst formulation, Pt/TiO₂ + Y, outperformed most reported catalysts in terms of superior low-temperature H₂-SCR activity, high N₂ selectivity, and broad operational temperature windows (please refer to Supplementary Table 1 for a performance comparison with reported catalysts). In addition, our research reveals that adding Y to Pt/TiO₂ effectively facilitated the capture of *in-situ* generated water, creating a water-rich environment around the Pt active sites. This environment led to weakened NO adsorption and enhanced H₂ activation, as confirmed by both experimental and theoretical studies. As a result, the Pt/TiO₂ + Y catalyst exhibited significantly improved H₂-SCR activity and N₂ selectivity. Meanwhile, this work advances the understanding that, in heterogeneous catalysis, tuning local reaction environments around active sites is crucial for improving catalytic performance, in addition to modifying the active sites themselves.

Significant research has been conducted in recent years to investigate the correlation between the properties and structure of H₂-SCR. Depending on the chemical promoter and support composition, Pd-based catalysts in stationary NO_x control applications may reach N₂-selectivity and NO_x conversion of 80-95% (Pt-based catalysts exhibited lower values). Different Pd-based catalyst compositions yield over 85% NO_x conversion in mobile applications. Authors can find a systematic review on “Innovative catalysts for the selective catalytic reduction of NO_x with H₂” (Farhan et al. *Fuel* 355 (2024) 129364), covering different types of catalysts, the significant impact of catalyst chemical components, including the active component (comprising noble and non-noble metals, and bimetallic compositions), supports (including zeolites and titania). The study also provides a comprehensive discussion on the effect of exhaust gas composition, including O₂, CO, CO₂, H₂O,...

Response: Thank you for your comments. The current major challenge in H₂-SCR research is developing a catalyst that demonstrates superior activity and high N₂ selectivity at low temperatures, while also maintaining broad operation temperature windows (*Ind. Eng. Chem. Res.* 2021, 60, 6613-6626; *Fuel* 2024, 355, 129364). Typically, Pd catalysts exhibit high H₂-SCR activity and N₂ selectivity at elevated temperatures, achieving N₂ selectivity and NO_x conversion rates of 80-95%. However, they show minimal activity at low temperatures, such as below 125 °C (*Catal. Today* 1998, 45, 135-138; *Chin. J. Catal.* 2010, 31, 261-263). In contrast, Pt catalysts perform well at low temperatures, but this advantage is offset by low N₂ selectivity and narrow operation temperature windows (*Chem. Eng. J.* 2015, 260, 419-426; *Chin. J. Catal.* 2015, 36, 197-203). In this work, the Pt/TiO₂ + Y catalyst system we developed showed superior catalytic performance, achieving NO_x conversion above 59% and N₂ selectivity above 58% at temperatures below 250 °C, even under the testing conditions with H₂O and CO₂. The broad operation temperature range (100-250 °C) combined with excellent catalytic performance makes the Pt/TiO₂ + Y system highly desirable for practical H₂-SCR applications (*Energies* 2021, 14, 8166). For better comparison with recently reported H₂-SCR catalysts in the literature, the reaction rates and N₂ selectivities at 100 and 200 °C were calculated and listed in Supplementary Table 1. Notably, the Pt/TiO₂ + Y system outperformed most reported Pt and Pd catalysts in terms of superior low-temperature activity and broad operation temperature windows, including the H₂-SCR catalysts highlighted in the mentioned review paper (*Fuel* 2024, 355, 129364).

Modification: Top/Page 5: In addition, this Pt/TiO₂ + Y catalyst showed much higher reaction rates and N₂ selectivity at 100 and 200 °C compared to most reported Pt and Pd catalysts (**Supplementary Table 1**).

Zeolites offering good thermal stability have been used for NH₃-SCR, so that several groups suggested zeolitic and zeolite-related systems also in the field of H₂-SCR. Borchers et al (*Top. Catal.* doi: 10.1007/s11244-022-01723-1) include TiO₂ and Y zeolite in a single Pd-based catalyst as 1%Pd/20%TiO₂/HY, which outperforms one active and selective benchmark catalyst for H₂-SCR. I am wondering if a similar formulation with Pt (same components than in your sample, but integrated in a single solid) will run under same reaction mechanism achieving worse/similar/better activity and selectivity. That is to say, the physical mixing Pt/TiO₂ + Y or Pt/TiO₂/Y, which would be preferred? Which will be better to modulate activity, selectivity and durability of the H₂-SCR catalyst? Furthermore, significant attention has been given to the role of noble metals at various sites within the zeolite structure, resulting in a thorough understanding of their effects. Inclusion of this discussion will greatly enhance the scientific insights of the paper.

Response: Thank you for your valuable suggestions. Following the same preparation method as for 1% Pd/20% TiO₂/HY, we prepared a 1% Pt/50% TiO₂/Y catalyst, referred as Pt/TiO₂/Y, and tested it for H₂-SCR reaction. As detailed in our response and modifications to Comment 4 from Reviewer #2, this Pt/TiO₂/Y catalyst, which features stronger interactions between Pt, TiO₂, and Y, showed lower H₂-SCR activity and N₂ selectivity compared to the physically mixed Pt/TiO₂ + Y. These results suggest that appropriate contact, rather than the stronger interactions between Pt/TiO₂ and Y, was beneficial for the H₂-SCR performance. Detailed structural characterization, including EDS mapping analysis of Pt/TiO₂ + Y before and after reaction (please see details in our response and modifications to your specific Comments 3 and 4), confirmed that there was no direct contact between Pt and Y. Additionally, as shown in Supplementary Fig. 5, the Pt/Y catalyst exhibited much lower activity compared to Pt/TiO₂, indicating that the Pt active sites on Y were less active than those on TiO₂. Therefore, the promotion effect by Y was not due to the deposition of Pt onto different sites within Y zeolite.

On the other, the mixing strategy of solids is not new in catalysis for many processes. The mixing of two complementary catalysts is generally made when looking for catalytic synergism (see the concept in Cang and Phillips, *Langmuir* 12 (1996) 2756). On the other hand, the approach given in the current manuscript is used when upgrading industrial catalytic processes without modifying the catalyst itself (Pt/TiO₂ in this case by mixing with commercial Y zeolite), also reported e.g. Fang et al., *Science* 377 (2022) 406.

Response: Thank you for your comments. We are confident that physically mixing Pt/oxide with zeolite to create a water-rich reaction environment around Pt sites is a novel concept, reported here for the first time in the context of H₂-SCR applications. In the study published in *Langmuir* (1996, 12, 2756), the authors explored the physical mixtures of FeCe/Grafoil and Pt/Grafoil (or Pd/Grafoil) for efficient 1-butene hydroisomerization. In such mixtures, both components were active and demonstrated synergistic effects, thereby enhancing the reaction. In contrast, in our system, the zeolite Y is inactive, with only Pt/TiO₂ serving as the active component. The improved catalytic performance is not due to synergism between the components, but rather to the enrichment of local water environments around the Pt active sites in Pt/TiO₂ by the presence of Y zeolite. Furthermore, this approach is conceptually distinct from the work published in *Science* (2022, 377, 406), which involved physically mixing the CoMnC catalyst with hydrophobic poly(divinylbenzene) to manage the water product in syngas conversion, resulting in a higher proportion of free surface area and thus an increased reaction rate. Our work stands out for its significant novelty in terms of catalytic materials, reactions, and the underlying concepts for enhancing catalytic performance, clearly differentiating it from the aforementioned studies. Hope this well addresses your concern.

In the following, I am suggesting some other considerations, in case they could help to authors for improving the manuscript.

1. In SCR metal-based catalysts supported on zeolites, the metal speciation and situation at various positions in the structure of zeolite is a key point to understand mechanism and behavior. This can be checked in many of the advanced catalysts designed summarized in paper by Farhan et al. (see above). What about a preparation with the Pt/TiO₂ supported on Y zeolite, instead of the physical mixing? If the idea is to surround Pt active sites with acidic sites of zeolite, a better intimate and stable contact should be achieved by deposition or incorporation of Pt/TiO₂ in the zeolite structure.

Response: Thank you for your constructive suggestions. Please see our detailed response and modifications to Comment 4 from Reviewer #2. We also addresses this point in the response to your prior comments.

2. The paper refers to selectivity to N₂ but does not analyse the rest of N-products, neither for Pt-TiO₂ catalyst nor for Pt/TiO₂ + Y mixing strategy. For example at 250 °C where selectivities are 17% and 58%, respectively (Figure 1).

Response: Thank you for your comments. Please see our detailed response and modifications to Comment 3 from Reviewer #2. The N-species distributions in different components during H₂-SCR on the Pt/TiO₂ and Pt/TiO₂ + Y catalysts have been added to this revised manuscript, based on your suggestion.

3. EDS mapping images revealed no contact between Pt and acidic sites after physical mixing. Authors should check and confirm if there is any modification in species distribution after reaction has been performed in order to confirm no real atomic contact between them during reaction conditions.

Response: Thank you for your kind suggestion. To determine whether there was any interaction between Pt and Y under reaction conditions, we performed EDS mapping analysis on the spent catalyst after the H₂-SCR reaction at 300 °C with H₂O, referred as (Pt/TiO₂ + Y)-*p*. As shown in Supplementary Fig. 16, the Pt species were well-aligned with the Ti distribution but did not correlate with the Si and Al distributions. This observation was consistent for both fresh and spent Pt/TiO₂ + Y samples, indicating that there was no direct contact between Pt and Y (acidic sites) during/after the reaction.

Modification: Top/Page 8: Additionally, the energy dispersive spectroscopy (EDS) mapping results of Pt/TiO₂ + Y revealed that the Pt/TiO₂ components were surrounded by Y zeolite particles, without obvious direct interaction between Pt species and Y, before and after H₂-SCR reaction (**Fig. 2c, Supplementary Fig. 16**).

Supplementary Fig. 16 | Structural characterization of Pt/TiO₂ + Y. EDS mapping images for (a) Pt/TiO₂ + Y and (b) (Pt/TiO₂ + Y)-*p* samples. The sample suffixed with “-*p*” represents the sample after reaction at 300 °C under testing conditions with H₂O.

4. Additionally, characterization shown in Table 1 should also be performed on used catalysts in order to discard any possible effect over surface properties.

Response: Thank you for your constructive suggestions. We have conducted XRD and N₂ adsorption-desorption experiments on the spent catalysts after the H₂-SCR reaction at 300 °C under testing conditions with H₂O, referred as Pt/TiO₂-*p* and (Pt/TiO₂ + Y)-*p*. As shown in the Supplementary Figs. 9 and 10, and Supplementary Table 2, there were no significant changes in the crystal structure or textural properties of Pt/TiO₂-*p* and (Pt/TiO₂ + Y)-*p* compared to their as-prepared counterparts. These results indicate that the catalysts were structurally stable under the reaction conditions.

Modification: Bottom/Page 6: Additionally, the Pt/TiO₂ + Y system demonstrated structural stability, with no apparent changes in crystal structure or textural properties after reaction at 300 °C under testing conditions with H₂O.

Supplementary Fig. 9 | Crystal structure. XRD patterns of Pt/TiO₂, Pt/TiO₂-*p*, Pt/TiO₂ + Y, (Pt/TiO₂ + Y)-*p*, and Y samples. The samples suffixed with “-*p*” represent the sample after reaction at 300 °C under testing conditions with H₂O.

Supplementary Fig. 10 | Porosity property. (a) N₂ adsorption-desorption isotherms, (b) pore size distributions for Pt/TiO₂, Pt/TiO₂-p, Pt/TiO₂ + Y, (Pt/TiO₂ + Y)-p, and Y samples. The samples suffixed with “-p” represent the sample after reaction at 300 °C under testing conditions with H₂O.

Supplementary Table 2 | TiO₂ grain size, BET surface area, pore volume, and average pore diameter for Pt/TiO₂, Pt/TiO₂-p, Pt/TiO₂ + Y, (Pt/TiO₂ + Y)-p, and Y samples.

Sample	TiO ₂ grain size ^a (nm)	BET surface area ^b (m ² /g)	Pore volume (cm ³ /g)		Average micropore/mesopore diameter (nm)
			Micropore ^c	Total ^d	
Pt/TiO ₂	20.0	81	0.032	0.178	-/7.1 ^d
Pt/TiO ₂ -p ^e	19.9	74	0.026	0.233	-/6.8 ^d
Pt/TiO ₂ + Y	20.6	349	0.164	0.318	0.6 ^c /3.8 ^d
(Pt/TiO ₂ + Y)-p ^e	20.3	325	0.150	0.350	0.6 ^c /3.8 ^d
Y	-	709	0.329	0.513	0.6 ^c /3.8 ^d

^a Determined according to the Scherrer equation using the full width at half maximum (FWHM) of the (101) peak of TiO₂.

^b Calculated using the Brunauer-Emmett-Teller (BET) method.

^c Determined using the Horvath-Kawazoe (HK) method.

^d Calculated using the non-local density functional theory (DFT) method.

^e The samples suffixed with “-p” represent the sample after reaction at 300 °C under testing conditions with H₂O.

5. Authors should explain the negative impact of adding increasing amounts of H-Y above 50%. One would expect a saturation effect in the improvement, rather than a NO_x conversion loss.

Response: Thank you for your comments and suggestion. In the selective catalytic reduction (SCR) of NO, the reducing agent (*e.g.*, NH₃ or H₂) can be oxidized by both NO and O₂ molecules. Enhanced activation of the reductant generally leads to improved low-temperature activity for both

NO and O₂ oxidation. However, at higher temperatures, this increased reductant activation can decrease the NO reduction activity, due to the more efficient oxidation of reductant by O₂ (with O₂ showing significantly higher concentration than NO under typical SCR conditions). In our study, we observed an increase in the low-temperature activity but a decrease in the high-temperature activity (Supplementary Fig. 4) for the Pt/TiO₂ + Y system as the H-Y content increased from 50% to 67%. Consistent with other reports (*Appl. Catal. B: Environ.* 2016, 188, 189-197; *Chem. Eng. J.* 2019, 355, 470-477; *Chem. Eng. J.* 2014, 252, 288-297), this effect could be attributed to the enhanced H₂ activation, as the introduction of Y has been shown to improve H₂ activation on Pt/TiO₂. Our tests further confirmed that the H₂ oxidation activity increased with the rising Y content in the Pt/TiO₂ + Y system (please see Supplementary Fig. 18), achieving 100% H₂ conversion on (Pt/TiO₂ + Y)-67% at temperatures above 40 °C.

Modification: Bottom/Page 8: The Pt/TiO₂ + Y system demonstrated superior H₂ oxidation activity compared to the Pt/TiO₂ reference, with the H₂ oxidation activity promoted further as the Y content in the Pt/TiO₂ + Y system increased (**Supplementary Fig. 18**). This additional increase in H₂ activation could improve the low-temperature activity and reduce the high-temperature activity, as observed on Pt/TiO₂ + Y-67% compared to that on Pt/TiO₂ + Y-50% (**Supplementary Fig. 4**).^{14,25}

Supplementary Fig. 18 | H₂ oxidation performance. H₂ oxidation activity on Pt/TiO₂, Pt/TiO₂ + Y-50%, and Pt/TiO₂ + Y-67% catalysts.

6. Please, specify if the correct Si/Al ratio for H-Y is 29 or 30, both values appear along the manuscript.

Response: We apologize for the typo. The SiO₂/Al₂O₃ ratio for H-Y used in this work was 30. This has been corrected in the revised manuscript. Thank you for your kind reminding.

Modification: Top/Page 5: When physically mixing the Pt/TiO₂ catalyst with an inactive commercial H-Y zeolite (SiO₂/Al₂O₃ molar ratio = 30)...

7. Please, could you suggest the final De-NO_x mechanism derived from your testing experiments? How is affecting the physical mixing to the mechanism with single Pt/TiO₂ catalyst? Is it more related to a change in composition of NO_x stream accessing to Pt/TiO₂ active sites, not related to the surface of the catalyst?

Response: Thank you for your question. As illustrated in Supplementary Fig. 17, on Pt/TiO₂ and Pt/TiO₂ + Y catalysts, the NO reaction orders were determined as 0.95 and 1.09, respectively, when the NO concentration was below 250 ppm (25.3 Pa), or 0.63 and 0.80, respectively, when the NO concentration was above 250 ppm; the H₂ reaction orders were determined as 0.48 and 0.32; the O₂ reaction orders were determined as -0.13 and -0.08. Under the specific conditions used in this study (500 ppm NO, 1% H₂, and 10% O₂), the H₂-SCR reaction rates can be expressed as: $r_{(\text{Pt/TiO}_2)} = k_1 \cdot [\text{NO}]^{0.63} \cdot [\text{H}_2]^{0.48} \cdot [\text{O}_2]^{-0.13}$ and $r_{(\text{Pt/TiO}_2 + \text{Y})} = k_2 \cdot [\text{NO}]^{0.80} \cdot [\text{H}_2]^{0.32} \cdot [\text{O}_2]^{-0.08}$, where k_1 and k_2 are constants. It is noteworthy that the NO and H₂ reaction orders on both Pt/TiO₂ (0.63 and 0.48, respectively) and Pt/TiO₂ + Y (0.80 and 0.32, respectively) are lower than 1. This can be attributed to the H₂-SCR reaction on both catalysts involving adsorbed NO and dissociated H* species, following the Langmuir-Hinshelwood (L-H) mechanism. Although the physical mixing with zeolite Y did not change the L-H mechanism on Pt/TiO₂, it significantly increased the NO reaction order (from 0.63 to 0.80) and noticeably decreased the H₂ reaction order (from 0.48 to 0.32) on the Pt/TiO₂ + Y catalyst. These changes indicate that the presence of Y reduced NO adsorption and simultaneously enhanced H₂ activation, mainly due to the increased H₂ coverage on the catalyst surface. These findings were systematically verified through various methods, including *in situ* DRIFTS, NO-TPD, H₂ oxidation, transient H₂-SCR reaction experiments, and DFT simulations. In addition, the results indicate that the presence of Y enriched the local water environment around the Pt sites, facilitating the formation of H₂O-covered Pt surface. This surface showed reduced NO adsorption and enhanced H₂ activation, promoting the H₂-SCR reaction accordingly. While the local H₂O-rich environment around the Pt sites could inhibit the diffusion of NO to the Pt surface, under steady testing conditions, the composition of the NO_x stream should remain unchanged. In other words, the *in situ* formation of H₂O-covered Pt surface in the Pt/TiO₂ + Y system was the main reason for the promoted H₂-SCR activity.

Modification: Bottom/Page 8: Under our testing conditions (500 ppm NO, 1% H₂, and 10% O₂), the H₂-SCR reaction rates can be expressed as: $r_{(\text{Pt/TiO}_2)} = k_1 \cdot [\text{NO}]^{0.63} \cdot [\text{H}_2]^{0.48} \cdot [\text{O}_2]^{-0.13}$ and $r_{(\text{Pt/TiO}_2 + \text{Y})} = k_2 \cdot [\text{NO}]^{0.80} \cdot [\text{H}_2]^{0.32} \cdot [\text{O}_2]^{-0.08}$, where k_1 and k_2 are constants. Notably, the NO and H₂ reaction orders on both Pt/TiO₂ (0.63 and 0.48, respectively) and Pt/TiO₂ + Y (0.80 and 0.32, respectively) are lower than 1. This suggests that the H₂-SCR reaction on both catalysts involved adsorbed NO and dissociated H* species, following the Langmuir-Hinshelwood (L-H) mechanism. Without changing the L-H mechanism, the enhanced H₂ activation could contribute to the improved H₂-SCR activity of Pt/TiO₂ + Y.

Top/Page 11: The physical mixing of Pt/TiO₂ with Y could further promote the adsorption of *in situ* generated H₂O (**Fig. 3e** and **f**), creating a H₂O-rich environment around the Pt sites and facilitating the formation of a H₂O-covered Pt surface. This surface could reduce the NO coverage on Pt sites, thereby improving H₂ activation and H₂-SCR performance (**Fig. 4a**).

REVIEWERS' COMMENTS

Reviewer #1 (Remarks to the Author):

In this article, authors investigated an interesting effect by trapped water in Y zeolite for H₂-SCR on Pt/TiO₂. By simply mixing Pt/TiO₂ and Y zeolite, a better H₂-SCR and N₂ selectivity can be achieved. In addition to this novel finding, authors thoroughly studied the kinetics of the reactions and further confirmed the observed effect and the mechanism.

Authors have addressed reviewers' comments accordingly and have improved the paper substantially. The paper was well written and is recommended to be accepted.

Reviewer #3 (Remarks to the Author):

Comments to paragraphs 1 and 2

In our previous analysis, we recognized a well-structured work, with well-designed experiments and discussion conducting to relevant results, by using the strategy of preparing a physical mixing of Pt/TiO₂ with commercial Y zeolite for NO_x removal in H₂-SCR applications. Our view was supported on lack of novelty in the concept of the strategy of mixing materials (catalysts and adsorbent), as already reported for enhancement of activity/selectivity in different catalytic processes, as referred in references indicated in our review report. The last was our main argument to suggest publication in a dedicated catalysis magazine better than in Nature Communications.

Based on the authors' rebutals we can accept novelty in applying the physical mixing strategy to the H₂-SCR. The authors have included the new Supplementary Table 1 for a performance comparison of reported catalysts, based on reaction rate (mmol/gmetal s) and N₂ selectivity (%). Although reaction rate referred to gram of catalyst is valid to compare behaviour of catalysts with different metal loading, it is not easily calculated from reported data achieved under different experimental conditions. To understand better comparison of low-temperature behaviour of H₂-SCR catalysts we suggest the T50 and T90 values of reported catalysts, which should be included in Supplementary Table 1. Selectivity is suitable to compare the product distribution (N-species distribution has now been included as Supplementary Fig. 1); however, selectivity to nitrogen in Supplementary Table 1, the Pt/TiO₂ + Y formulation does not show superior behaviour than other reported catalysts.

Rest of comments

Concerning the rest of our previous general and specific comments, we think that the authors have adequately addressed in the revised version of the manuscript. Therefore, we have not further comments.

Reviewer #4 (Remarks to the Author):

I co-reviewed this manuscript with one of the reviewers who provided the listed reports.

This is part of the Nature Communications initiative to facilitate training in peer review and to provide appropriate recognition for Early Career Researchers who co-review manuscripts.

Notes:

According to the comments and suggestions from the reviewers, the Main Manuscript and the Supplementary Information have been carefully revised. All the changes are indicated using **red fonts** for the reviewers' convenience.

Point-to-Point Response to the Reviewers' Comments:

Reviewer #1 (Remarks to the Author):

In this article, authors investigated an interesting effect by trapped water in Y zeolite for H₂-SCR on Pt/TiO₂. By simply mixing Pt/TiO₂ and Y zeolite, a better H₂-SCR and N₂ selectivity can be achieved. In addition to this novel finding, authors thoroughly studied the kinetics of the reactions and further confirmed the observed effect and the mechanism.

Authors have addressed reviewers' comments accordingly and have improved the paper substantially. The paper was well written and is recommended to be accepted.

Response: Thank you so much for your kind recommendation of publishing our manuscript on *Nature Communications*.

Reviewer #3 (Remarks to the Author):

Comments to paragraphs 1 and 2

In our previous analysis, we recognized a well-structured work, with well-designed experiments and discussion conducting to relevant results, by using the strategy of preparing a physical mixing of Pt/TiO₂ with commercial Y zeolite for NO_x removal in H₂-SCR applications. Our view was supported on lack of novelty in the concept of the strategy of mixing materials (catalysts and adsorbent), as already reported for enhancement of activity/selectivity in different catalytic processes, as referred in references indicated in our review report. The last was our main argument to suggest publication in a dedicated catalysis magazine better than in Nature Communications. Based on the authors' rebutals we can accept novelty in applying the physical mixing strategy to the H₂-SCR. The authors have included the new Supplementary Table 1 for a performance comparison of reported catalysts, based on reaction rate (mmol/gmetal s) and N₂ selectivity (%). Although reaction rate referred to gram of catalyst is valid to compare behaviour of catalysts with different metal loading, it is not easily calculated from reported data achieved under different experimental conditions. To understand better comparison of low-temperature behaviour of H₂-SCR catalysts we suggest the T50 and T90 values of reported catalysts, which should be included in Supplementary Table 1. Selectivity is suitable to compare the product distribution (N-species distribution has now been included as Supplementary Fig. 1); however, selectivity to nitrogen in Supplementary Table 1, the Pt/TiO₂ + Y formulation does not show superior behaviour than other reported catalysts.

Rest of comments

Concerning the rest of our previous general and specific comments, we think that the authors have adequately addressed in the revised version of the manuscript. Therefore, we have not further comments.

Response: We appreciate your recognition on the novelty of this manuscript now. Given that the catalysts have different metal loadings and have been tested under varied conditions, we believe that calculating the reaction rate relative to the mass of metal is the most suitable and equitable way to compare the catalysts, despite the complexity of the calculation. We agree with you that adding the T50 and T90 data will provide a more comprehensive comparison. Following your suggestion, these data have been included in Supplementary Table 1. Please see details in the revised Supplementary Information.

While the N₂ selectivity on Pt/TiO₂ + Y does not exceed that on reported catalysts, particularly Pd-based ones, its exceptional low-temperature activity and broad operation temperature window make it superior to most of the reported catalysts, showing great potential for practical application.

Reviewer #4 (Remarks to the Author):

Response: Thank you very much for your comments and suggestions. For your reference, please see our detailed response and modifications to Comments from Reviewer #3.